# Similar neural and perceptual masking effects of low-power optogenetic stimulation in primate V1

**Spencer Chin-Yu Chen[1,2,3,4], Giacomo Benvenuti[2,3,4], Yuzhi Chen[2,3,4], Satwant Kumar[2,3,4], Charu Ramakrishnan[5], Karl Deisseroth[5,6,7,8,9], Wilson S Geisler[2,3,10], Eyal Seidemann[2,3,4,10]***

[1]Department of Neurosurgery, Rutgers University, New Brunswick, United States; [2]Center for Perceptual Systems, The University of Texas at Austin, Austin, United States; [3]Department of Psychology, University of Texas, Austin, United States; [4]Department of Neuroscience, University of Texas, Austin, United States; [5]CNC Program, Stanford University, Stanford, United States; [6]Department of Bioengineering, Stanford University, Stanford, United States; [7]Neurosciences Program, Stanford University, Stanford, United States; [8]Department of Psychiatry and Behavioral Sciences, Stanford University, Stanford, United States; [9]Howard Hughes Medical Institute, Stanford University, Stanford, United States; [10]Neurosciences Program, University of Texas, Austin, United States

**Abstract** Can direct stimulation of primate V1 substitute for a visual stimulus and mimic its perceptual effect? To address this question, we developed an optical-genetic toolkit to 'read' neural population responses using widefield calcium imaging, while simultaneously using optogenetics to 'write' neural responses into V1 of behaving macaques. We focused on the phenomenon of visual masking, where detection of a dim target is significantly reduced by a co-localized medium-brightness mask (Cornsweet and Pinsker, 1965; Whittle and Swanston, 1974). Using our toolkit, we tested whether V1 optogenetic stimulation can recapitulate the perceptual masking effect of a visual mask. We find that, similar to a visual mask, low-power optostimulation can significantly reduce visual detection sensitivity, that a sublinear interaction between visual- and optogenetic-evoked V1 responses could account for this perceptual effect, and that these neural and behavioral effects are spatially selective. Our toolkit and results open the door for further exploration of perceptual substitutions by direct stimulation of sensory cortex.

**\*For correspondence:**
eyal@austin.utexas.edu

**Competing interest:** The authors declare that no competing interests exist.

## Editor's evaluation

This work examined a long-standing technical and conceptual question in systems neuroscience: can artificial perturbation of primary sensory cortex (in this case V1) mimic the perceptual effects of natural sensory stimulation? This technically impressive work combined optogenetics and visual psychophysics in monkeys to show that certain controlled patterns of V1 simulation can recapitulate a relatively simple visual perceptual effect involving visual masking. The results provide a proof-of-concept for a new set of approaches for studying the neural basis of visual perception.

## Introduction

A central goal of sensory neuroscience, and a prerequisite for the development of effective sensory cortical neuroprostheses, is to understand the nature of the neural code – that is, to determine which

patterns of neural activity in early sensory cortex are necessary and sufficient to elicit a given percept. A promising experimental paradigm for testing specific hypotheses regarding the neural code is to use optogenetic stimulation (optostim) to directly insert neural signals into sensory cortex of alert, behaving animals. By using simultaneous optical imaging of genetically encoded calcium indicators, one can measure the impact of these inserted signals on cortical circuits, calibrate the evoked neural population responses, and compare them to those evoked by sensory stimuli. Finally, by carefully monitoring behavior while animals perform demanding sensory tasks, the perceptual consequences of these inserted signals can be assessed and compared with theoretical predictions.

This powerful experimental approach for all-optical interrogation of sensory cortex has recently been successfully used to study the neural code in rodents (e.g., *Marshel et al., 2019*; *Carrillo-Reid et al., 2019*; *Dalgleish et al., 2020*). However, the merging of optostim and optical imaging of calcium signals in behaving non-human primates (NHPs), an important animal model for studying human perception, is still in its infancy (*Ju et al., 2018*), and simultaneous imaging, optogenetic stimulation, and behavior have not been achieved previously in NHPs. Multiple barriers remain before this approach can be used routinely in NHPs (*Tremblay et al., 2020*; *El-Shamayleh and Horwitz, 2019*; *Galvan et al., 2017*). For example, while optostim in early sensory cortex has been demonstrated to cause clear neural (e.g., *Ju et al., 2018*; *Ruiz et al., 2013*; *Nassi et al., 2015*; *Jazayeri et al., 2012*; *May et al., 2014*; *Andrei et al., 2019*) and behavioral (e.g., *Ju et al., 2018*; *Jazayeri et al., 2012*; *May et al., 2014*; *Andrei et al., 2019*) effects in NHPs, such experiments typically require high-power optostimulation (10s to 100s of mW/mm$^2$), and the evoked neural responses are typically monitored by electrophysiology which only captures a small fraction of the optostim-triggered neural population response.

To overcome some of these barriers, we have been working to develop a bi-directional optical-genetic toolkit for measuring ('reading') and modulating ('writing') neural population responses in the cortex of behaving macaques (*Figure 1A*). First, we developed viral-based methods for stable long-term co-expression of a calcium indicator (GCaMP6f; *Chen et al., 2013*) and a red-shifted opsin (C1V1; *Packer et al., 2012*) in a population of excitatory V1 neurons (*Figure 2A*). Second, we developed a widefield imaging system that allows us to simultaneously optostimulate and image NHP cortex with negligible cross-talk between these optical read/write components (*Figure 2B*).

Our long-term goal is to use this toolkit to study the nature of the neural code by testing the extent to which one can substitute direct optostim for visual stimulation in a behaving NHP that is engaged in a demanding perceptual task. Previous pioneering studies using electrical microstimulation in area MT of macaques performing a random-dot direction discrimination task demonstrated that microstimulation at the center of a direction selective MT column can substitute for a visual motion signal and strongly bias perceptual reports in favor of the direction preferred by the stimulated neurons (*Salzman et al., 1990*; *Salzman et al., 1992*). An important feature of these experiments was that monkeys were always rewarded for reporting the direction of motion of the visual stimulus irrespective of the microstimulation. The monkeys' reports were consistently biased in the direction preferred by the stimulated neurons, even though the monkeys were not rewarded based on the microstimulation. This strongly suggests that the perceptual consequences of MT microstimulation are similar to the perceptual effects of a visual stimulus moving in the preferred direction. Our goal here was to use a similar reward strategy, and test whether optostim can be used to affect perception in a predictable way while monkeys perform a challenging visual task and are rewarded solely based on the visual stimulus.

As a first step in using our toolkit, we focused on the phenomenon of visual masking because it provides a simple test of perceptual substitution. The response of neurons in the visual cortex to visual stimuli is highly nonlinear. For example, the response to a stimulus that is at 100% contrast is typically much lower than twice the response at 50% contrast (*Albrecht and Hamilton, 1982*). A perceptual correlate of this neural nonlinearity is visual masking. The threshold for detecting a dim visual target is low on a uniform gray background (*Figure 1B*, top) and significantly higher when it is added to a co-localized medium-brightness pedestal mask (*Figure 1B*, bottom), consistent with Weber's law (*Cornsweet and Pinsker, 1965*).

We used our toolkit to ask three related questions. (1) Can we substitute for a visual mask with low-power (<1 mW/mm$^2$) direct optostim of the visual cortex and cause significant elevation in detection threshold of visual targets? (2) If so, is there a sublinear interaction between visual- and optogenetic-evoked neural responses in behaving macaque V1 (similar to the sublinearity between a visual mask

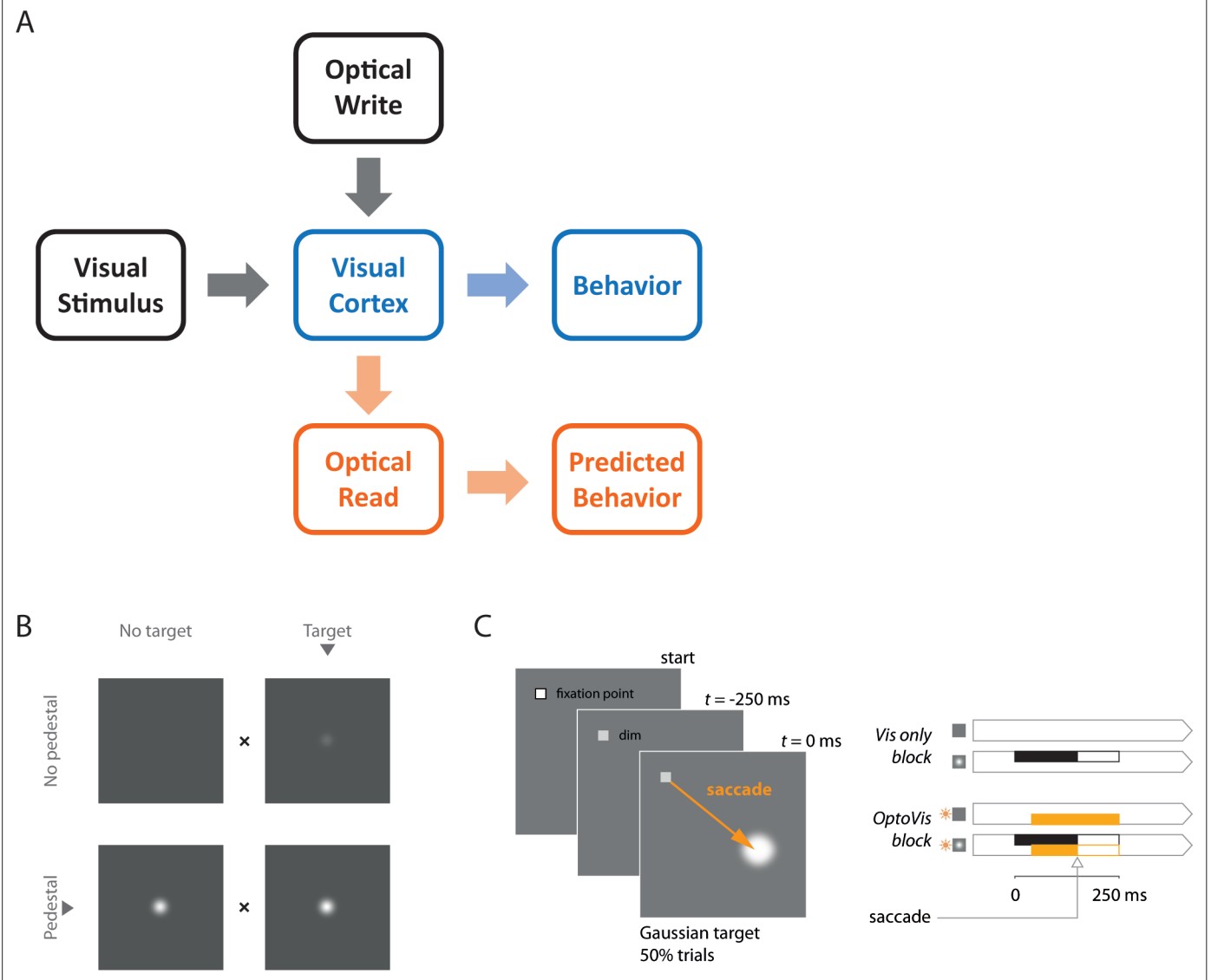

**Figure 1.** A general framework for bi-directional optical-genetic probing of the visual cortex in behaving monkeys, a demonstration of visual masking, and a detection task for probing the masking effect of V1 optostimulation. (**A**) General framework. Our optical-genetic toolkit allows us to provide the subject with two types of inputs either separately or in combination: (**i**) visual ('visual stimulus') and (**ii**) direct optostimulation ('optical write'). At the same time, we have access to two outputs: (**i**) neural responses measured by widefield optical imaging ('optical read') and (**ii**) behavioral responses ('behavior'). Imaging allows us to measure the neural impact of the inserted signals, calibrate the evoked neural population responses, and compare them to those evoked by sensory stimuli. Finally, the neural and perceptual consequences of these inserted signals can be assessed and compared with theoretical predictions. Here, we use this toolkit to measure the interactions between visual and direct optostimulation in macaque V1. (**B**) Demonstration of visual masking. When fixating on the 'x' between the pair of panels, a visual target (a dim white Gaussian) can be easily detected when added to a uniform gray background (top), but is much harder to detect when added to a Gaussian pedestal mask (bottom), a phenomenon known as luminance masking (***Cornsweet and Pinsker, 1965***; ***Whittle and Swanston, 1974***). The goal of the current study was to determine whether direct optogenetic stimulation of V1 can substitute for a visual mask and elevate detection threshold of a visual target. Note that in the actual behavioral task, only a single stimulus was presented in one hemifield. The animal had to distinguish between 'target' and 'no-target' and conditions with 'pedestal' and with 'no pedestal' were run in separate blocks (see panel C). (**C**) The behavioral task adopted to quantify the masking effects of optostimulation. Two monkeys were trained to detect a small white Gaussian target that appeared at a known location 250 ms after a temporal cue (dimming of the fixation point) in half of the trials. The monkeys indicated target absence by maintaining fixation at the fixation point and target presence by shifting gaze to the target location as soon as it was detected. The visual target was present for a maximum of 250 ms and was terminated as soon as the monkey initiated a saccade. The optostim was initiated 40 ms after the expected time of target onset (to account for the latency of V1 responses) and was terminated together with the visual stimulus. Blocks of trials without optostim (right top) and with optostim (right bottom) were run separately. In optostim blocks, optostim (orange) was applied *on every trial*, acting as a substitute for the visual mask ('pedestal') in **B**. The monkeys were always rewarded based on the presence or absence of the visual target only (right panels, black), irrespective of the optostim condition (orange).

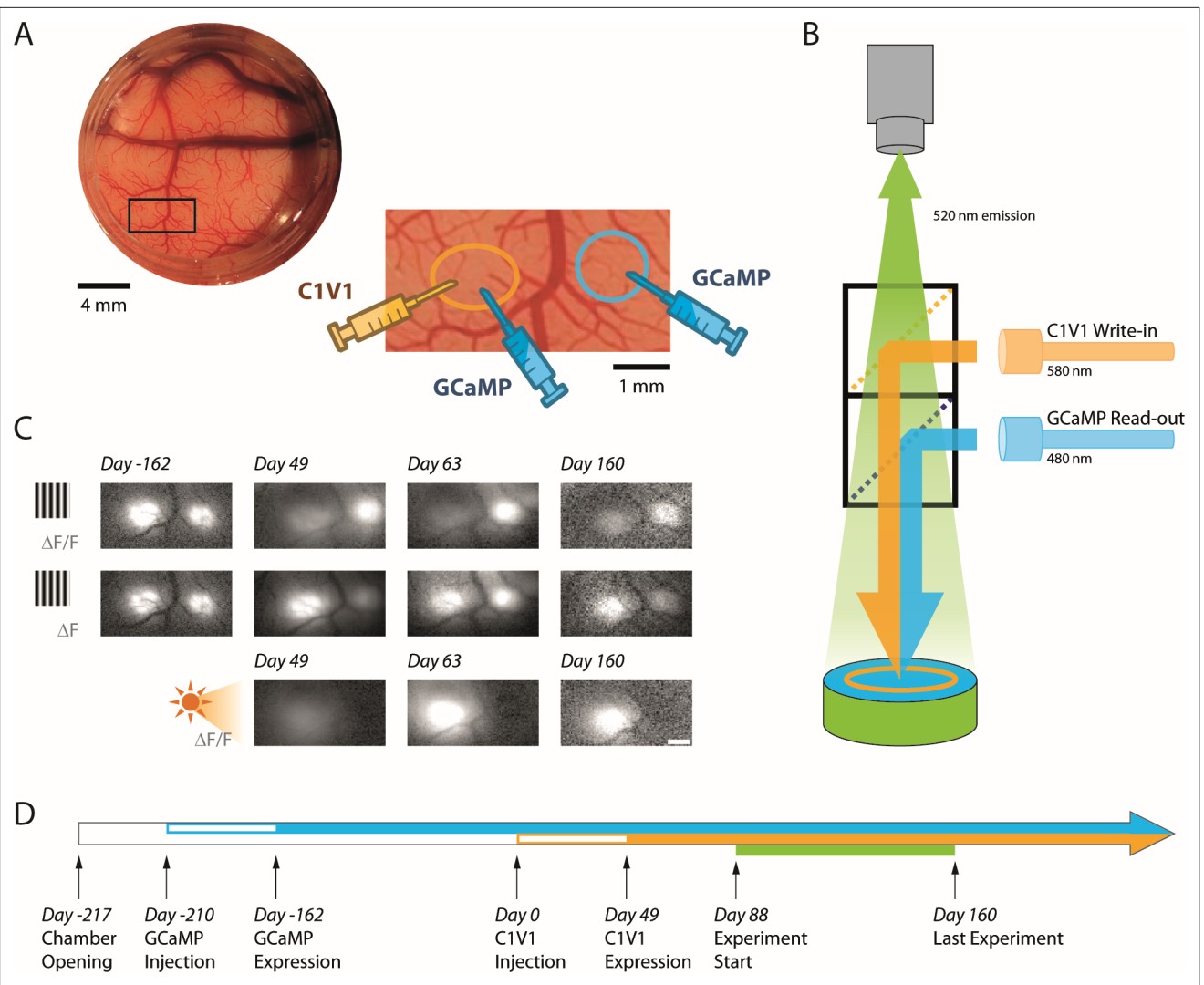

**Figure 2.** Simultaneous calcium imaging and optostim in V1 of monkeys performing a visual detection task. (**A**) Top left: Picture of cranial window over macaque V1 in Monkey L seen through the artificial dura, with a region of interest indicated by the black rectangle. Bottom right: The zoomed in region encompasses two nearby injection sites of viral vectors that are about 3 mm apart (~0.6° separation between the corresponding receptive fields). The vector carrying the transgene for the calcium indicator GCaMP6f was injected at both sites. The vector carrying the transgene for the opsin C1V1 was only injected at the site on the left after GCaMP expression was verified (panel C, top). We refer to the site with GCaMP and C1V1 as the 'C1V1-site' and the site with GCaMP only as the 'GCaMP-only' site. The centers of the receptive fields of the V1 neurons were 1.5° eccentricity (–35° from the right-hand horizontal meridian) at the C1V1 site and 2.0° eccentricity (–45° from the horizontal meridian) at the GCaMP-only site. (**B**) Schematic diagram of our combined imaging and optostim setup. To image the calcium indicator GCaMP6f signals, the cortex is illuminated through a dichroic mirror with blue light (480 nm). Green fluorescent signals reflecting increase in intracellular calcium concentration due to neural activity are collected by a sensitive sCMOS camera. To stimulate the red-shifted opsin C1V1, orange light (580 nm) is reflected to the cortex through a second dichroic mirror. The blue and orange lights are blocked from the camera by the dichroic mirrors and an emission filter so that the camera only collects the green fluorescent signals. Note that imaging was performed simultaneously at both sites and that light stimulation covered both sites. (**C**) Response maps for visual and optostim in Monkey L at different time points. A large visual grating evoked a GCaMP response at both sites (top and middle rows). When expressed as *ΔF/F* (top row), visual-evoked response at the C1V1 site was weaker following C1V1 expression due to increase baseline fluorescence from the eYFP that is attached to the C1V1 (see Materials and methods). When considering *ΔF* (middle row), the response evoked by the visual stimulus is comparable at the two sites. Optostim with stimulation light covering both locations elicits a strong GCaMP response at the C1V1 site and little or no response at the GCaMP-only site (bottom row). Scale bar 1 mm. (**D**) Timeline of GCaMP and C1V1 viral injections, first detection of expression, and behavioral experiments in Monkey L.

The online version of this article includes the following figure supplement(s) for figure 2:

**Figure supplement 1.** Stable expression of GCaMP and C1V1 in Monkey T.

**Figure supplement 2.** Timeline of neural and behavioral effects of optostimulation.

**Figure supplement 3.** Excessive blue GCaMP excitation light can affect behavioral performance.

and a visual target) that could account for the behavioral masking effect? (3) What is the spatial specificity of these effects, or in other words, how do the behavioral and neural effects depend on the location of the visual target relative to the receptive fields of the optostimulated neurons?

## Results

The overarching goal of the current study was to test two related hypotheses: (1) that low-power optogenetic stimulation (optostim) can substitute for a localized visual mask and cause a significant drop in behavioral detection sensitivities, and (2) that this perceptual effect is caused by a sublinear interaction between the neural responses elicited in macaque V1 by simultaneous visual and direct optogenetic stimulation. To test these hypotheses, we developed an optical-genetic toolkit that allowed us to simultaneously measure and stimulate optically neural responses in the cortex of behaving macaques.

### An optical-genetic toolkit for reading and writing neural population responses in behaving macaques

Our toolkit employs genetic and optical components. For the genetic component, we used methods we developed previously (*Seidemann et al., 2016*) to express in two macaques, at two V1 sites per monkey, a calcium indicator (GCaMP6f; *Chen et al., 2013*) in excitatory neurons (see Materials and methods). After we observed robust visual-evoked GCaMP responses (*Figure 2C–D*; *Figure 2—figure supplement 1*), we used a second viral vector to express the red-shifted opsin C1V1 in excitatory neurons at a one of the GCaMP expressing sites (*Figure 2A*) (see Materials and methods). We refer to the site with GCaMP and C1V1 as the 'C1V1 site' and the site with GCaMP only as the 'GCaMP-only site'. We find robust and stable co-expression of GCaMP and C1V1, which lasts for the lifetime of our imaging chambers (>2 years in both chambers; *Figure 2D*; *Figure 2—figure supplement 1*; *Figure 2—figure supplement 2*). The size of the area activated by optogenetic stimulation was ~2 $mm^2$ at half of the maximal response. This area contains several hundred thousand neurons (*Kelly and Hawken, 2017*).

Second, in order to image visual- and optogenetic-evoked GCaMP responses, we modified our widefield fluorescent imaging system to include an optical stimulation path (*Figure 2B*). Even though the stimulation light covered both sites, optostim-driven responses could only be measured at the C1V1 site (*Figure 2C,* bottom). This confirms that the optical stimulation light (at 580 nm) was efficiently blocked from contaminating our GCaMP imaging measurements (see Materials and methods). Using this system, we can measure responses to visual stimulation, to optostim, and to the combination of the two. Pilot imaging experiments revealed robust GCaMP responses to optostim at peak power densities below 1 $mW/mm^2$ that were within the ballpark of the visual-evoked responses. Therefore, for Monkey L, we chose to test the behavioral and neural effect of optostim at peak power levels of 0.6 and 1.2 $mW/mm^2$ (*Figure 2C*). To image the calcium signals we excited GCaMP molecules with blue light (at 480 nm). Pilot detection experiments revealed that, due to the broad excitation spectra of C1V1 (*Packer et al., 2012*), the blue GCaMP excitation light can affect the monkey's detection performance (*Figure 2—figure supplement 3*). We therefore minimized the blue light level to 0.05 $mW/mm^2$ in Monkey L and 0.01 $mW/mm^2$ in Monkey T by running the camera at a low frame rate (20 Hz) and at a very low saturation level (see Materials and methods). The monkeys' behavioral thresholds with the low-level blue light were comparable to their thresholds during training without blue light. Therefore, under the conditions tested here, the cross-talk between the optical 'read' and optical 'write' components in our system is negligible.

### Low-power optostim in macaque V1 causes a large and selective behavioral masking effect

To test the hypothesis that low-power optostim can substitute for a visual mask and cause a significant decrease in neural and behavioral detection sensitivities, we trained two macaque monkeys to perform a reaction-time visual detection task (*Figure 1C*). A visual target (a small white Gaussian stimulus; 0.33° FWHM) appeared at a known location on half of the trials. To receive a reward, the monkey had to indicate the presence of the target by making a saccade to the target location on target-present trials, or to maintain fixation on target-absent trials (see Materials and methods). In separate blocks, we measured the monkeys' ability to detect the visual target with no optostim (*Figure 1C*, top

right) and when optostim was applied *in all trials* (*Figure 1C*, bottom right). Note that in the optostim blocks, optostim acted as a visual mask (i.e., a co-localized baseline visual stimulus that appears on all trials; *Figure 1B*, bottom) and the monkey had to discriminate between trials with optostim only and trials in which the visual stimulus and optostim input coincided. Optostim was applied in 5 ms pulses at 44 Hz; optostim onset was delayed 40 ms relative to visual target onset to compensate for the latency of the visual responses so that response onset to the visual target and to optostim coincided; optostim offset matched visual target offset (*Figure 1C*, right; see Materials and methods).

If low-power optostim and visual stimuli activate separate neural populations in V1, or if they combine additively in the same population, then we might expect optostim to have little or no perceptual masking effects. On the other hand, if the responses to visual stimulation and optostim interact sublinearly in V1 (as do the responses to two superimposed visual stimuli; *Figure 1B*, bottom), we would expect low-power optostim to cause a significant increase in the monkeys' detection threshold relative to the no-optostim blocks. Consistent with the second hypothesis, we found that optostim caused a large perceptual masking effect that led to a reduction in the monkey's ability to detect the visual target in the presence of optostim (*Figure 3*). Across all experiments in Monkey L, optostim at 0.6 mW/mm$^2$ caused a large drop in the monkey's bias-adjusted accuracy (*Figure 3C*; see *Figure 3— figure supplement 1* for a non-bias-adjusted version of the figure) and a corresponding increase in detection threshold (*Figure 3D*). In this monkey, optostim caused primarily a large drop in hit rate (*Figure 3A*) and much smaller increase in false alarm rate (one minus correct rejections rate) (*Figure 3B*). Increasing the optostim peak power density to 1.2 mW/mm$^2$ caused a further decrease in accuracy and an increase in detection threshold. In addition to a significant increase in detection threshold, optostim also caused a significant reduction in the monkey's detection criterion at the higher power density (*Figure 3E*). This could reflect incomplete adjustment of the internal criterion used by the monkey for reporting the presence of the target. In other words, the results suggest that the monkey's criterion adjustment was insufficient to fully compensate for the perceptual consequences of the optostim.

A similar large perceptual masking effect of optostim was observed in Monkey T (*Figure 3F–J*). Because the behavioral effects of optostim at 0.6 mW/mm$^2$ were already high, we also tested the effect of optostim at 0.3 mW/mm$^2$. Even at this low-power level, we observed a significant increase in detection threshold in the presence of optostim. In this monkey, optostim caused mainly an increase in the false alarm rate and a smaller drop in hit rate, and a significant reduction in the monkey's detection criterion. Thus, our results reveal that low-power optostim in macaque V1 can substitute for a visual mask and cause large behavioral masking effects.

In contrast to the large and significant effect of optostim on the monkey's detection performance, the effect of optostim on the monkey's reaction times were weak and variable (*Figure 3—figure supplement 2*). Similarly, we did not observe a clear effect of optostim on the end points of the monkeys' saccades toward the target (*Figure 3—figure supplement 3*).

If the effect of optostim is similar to the effect of a visual mask, we would expect the behavioral effect to be selective to visual targets that fall in the receptive field of the C1V1 expressing neurons. Alternatively, if the optostim acts as a general distracter, we may obtain a similar behavioral effect even when the target falls at nearby locations. We repeated the experiments while presenting the target at a location in the visual field that corresponds to the receptive field of the neurons at the GCaMP-only site (distance from visual field location corresponding to the C1V1 site: 0.6° in Monkey L and 0.7° in Monkey T). Despite the fact that optostim light covered both the C1V1 and the GCaMP-only sites, and that the receptive fields at the two locations partially overlapped (see Materials and methods), optostim had a much smaller effect when the target was presented at the visual field location corresponding to the GCaMP-only site (*Figure 4*). These results show that the behavioral effect of optostim is spatially selective, consistent with the hypothesis that optostim is acting as a visual mask rather than as a general distracter.

## Sublinear interactions between visual- and optostim-evoked neural responses in V1 can explain the behavioral masking effects

Our next goal was to use the 'reading' component of our optical-genetic toolkit to determine whether a neural correlate of the behavioral masking effect can be observed in macaque V1. Our previous work (*Seidemann et al., 2016*) demonstrates that widefield GCaMP signals, which measure the pooled

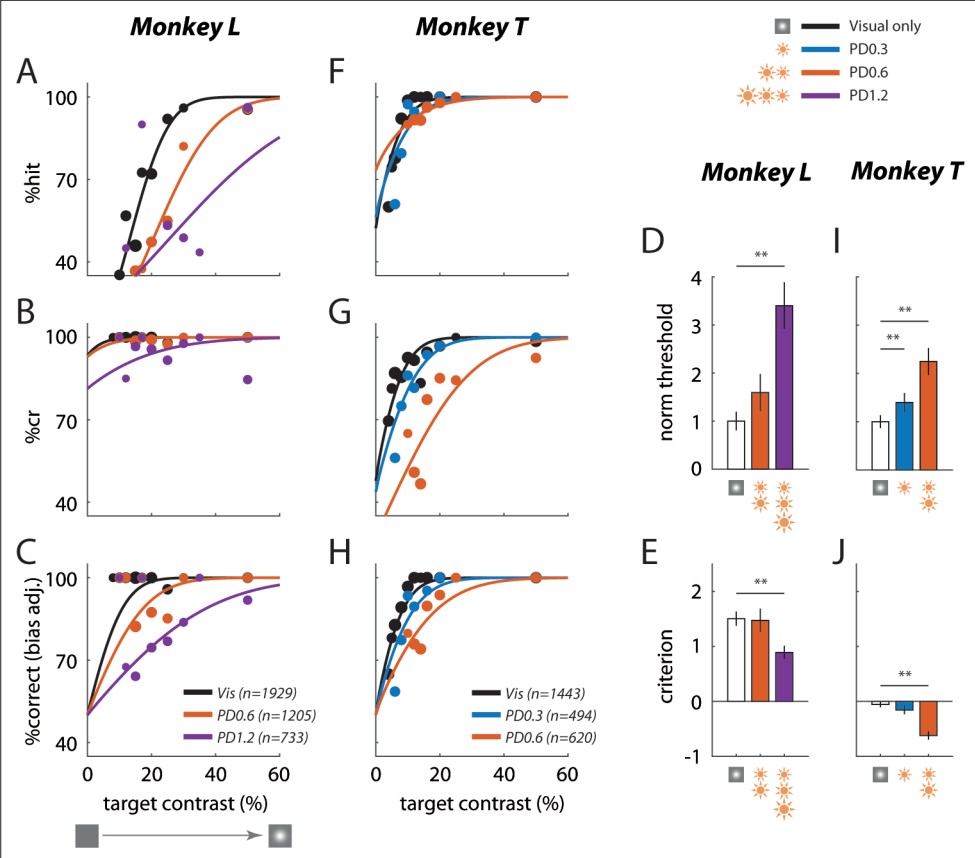

**Figure 3.** Low-power optogenetic stimulation leads to a large behavioral masking effect in a visual detection task. Summary of behavioral performance as a function of target Weber contrast at the C1V1 site for Monkey L (**A–E**) and Monkey T (**F–J**). Performance is broken down into: (**A**) '"hits' – target-present trials correctly identified, and (**B**) 'correct rejections' or 'cr' – target-absent trials correctly identified. (**C**) Detection performance corrected for the animal's decision bias (imbalance between hits and correct rejections; see Materials and methods for correction approach, and *Figure 3—figure supplement 1* for raw %correct data). The solid lines plot the fitted psychometric curves (see Materials and methods for fitting details). Data were pooled across all visual-only experiments and across all experiments of the same optostim peak power density (PD) level (in mW/mm²). The total number of trials pooled for each group is indicated in the legend, and the size of the plotted markers represents the relative number of trials executed at each target contrast. (**D**) Detection threshold with optostim (at 69% correct) plotted relative to the visual-target-only threshold. Mean thresholds and error bars were normalized by the mean visual threshold. (**E**) Criterion bias by each optostim power level. Criterion is reported in $d'$ units from the optimal criterion (criterion = 0), with positive values representing bias toward choosing target absent. Error bars in (**D**) and (**E**) indicate bootstrapped standard deviation (number of bootstrap runs = 1000) of the detection threshold and criterion, respectively. (**F**)–(**J**) are same as (**A**)–(**E**) for Monkey T's C1V1 site. Asterisks mark statistically significant changes from visual-only behavior (*p < 0.05; **p < 0.01; bootstrapped paired difference with Bonferroni correction for multiple comparisons).

The online version of this article includes the following figure supplement(s) for figure 3:

**Figure supplement 1.** Pre-bias-adjusted psychometric curves.

**Figure supplement 2.** Effect of optostim on reaction time.

**Figure supplement 3.** Effect of optostim on saccade end points.

---

activity of populations of neurons in a Gaussian shaped region with a space constant of ~0.2 mm (***Chen et al., 2012***), are approximately linearly related to the locally pooled spiking activity in macaque V1. Therefore, any sublinear interaction between visual- and optogenetic-evoked GCaMP responses is likely to reflect sublinear interaction at the level of V1 spiking activity. While the monkeys performed the detection task, we measured V1 GCaMP responses to the visual stimuli with and without simultaneous optostim. *Figure 5* shows results from two experiments in Monkey L when the target was

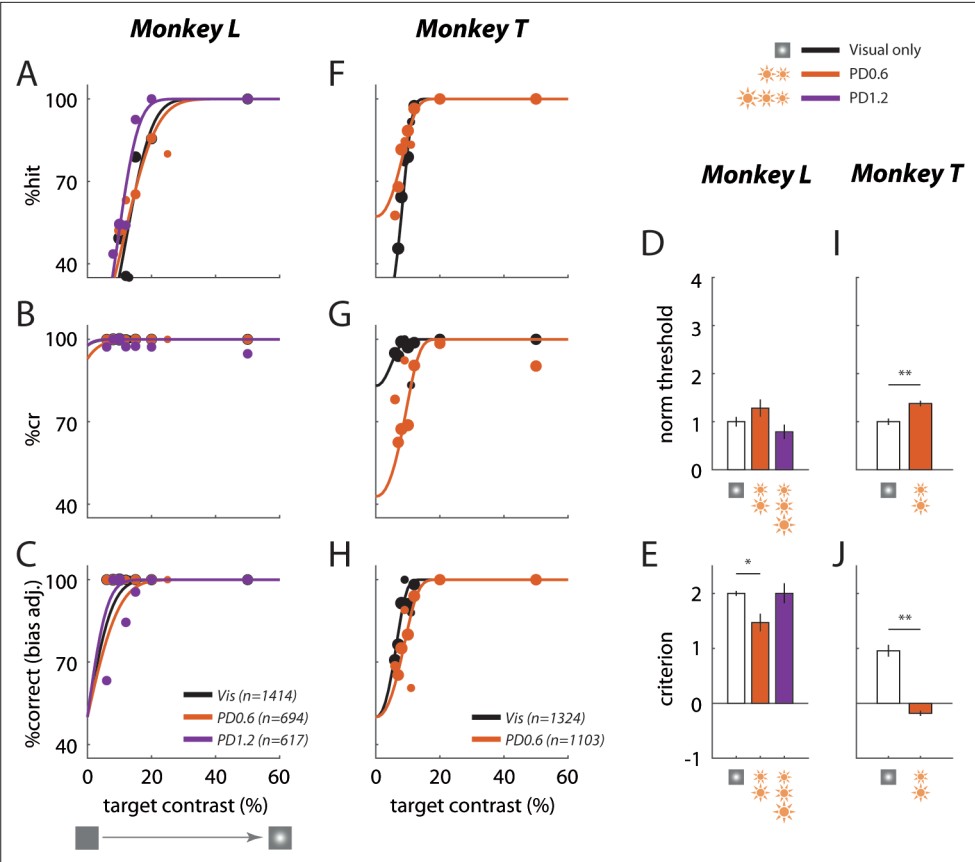

**Figure 4.** Behavioral effect of optostim is spatially selective. Same as *Figure 3* but visual target at location corresponding to the GCaMP-only site.

presented at the visual field location corresponding to the C1V1 site. In the first experiment, both the visual target (at 50% Weber contrast) and direct optogenetic stimulation produced similar amplitude responses (*Figure 5A and B*; the ratio of response to optostim and visual stim at 50% contrast was $R_{opto}/R_{vis50} = 0.95$). The response was stronger when the visual stimulus and the optostim mask were presented simultaneously (*Figure 5C*), but the visual-evoked response in the presence of the optostim mask was significantly reduced (*Figure 5D*), consistent with the hypothesis that these two signals interact sublinearly in V1. The time course of the responses reveals similar latency (note that optostim started 40 ms after visual stimulus onset), and a clear reduction in the visual-evoked response in the presence of optostim (*Figure 5E*; compare black curve to dashed orange curve). Finally, to measure the overall reduction in the visual-evoked responses in the presence of the optostim, we computed the average visual-evoked response, with and without optostim, at five target amplitudes (*Figure 5F*; see Materials and methods). Optostim caused a large reduction in the visual-evoked responses at all target amplitudes (average reduction of 50.4%).

In a second experiment conducted 6 weeks later, the optostim at the same power level produced a response that was much larger than the visual-evoked response (*Figure 5G*; $R_{opto}/R_{vis50} = 3.11$), suggesting a possible increase in the C1V1 expression level between these two experiments (see *Figure 2—figure supplement 2* for the time course of $R_{opto}/R_{vis50}$). Consistent with higher level of C1V1, the reduction of the visual-evoked response in the presence of optostim was also larger (*Figure 5H*), leading to almost complete elimination of the visual-evoked responses in the presence of the optostim mask. Similarly, the behavioral effect of optostim in the second experiment (64% increase in detection threshold) was larger than in the first experiment (35% increase in detection threshold).

A summary of the physiological results (*Figure 6*) reveals a strong masking effect of optostim on V1 responses, which is even stronger in Monkey T, where even the lowest power level of 0.3 mW/mm$^2$ almost abolished all visual-evoked GCaMP responses (*Figure 6C–D*).

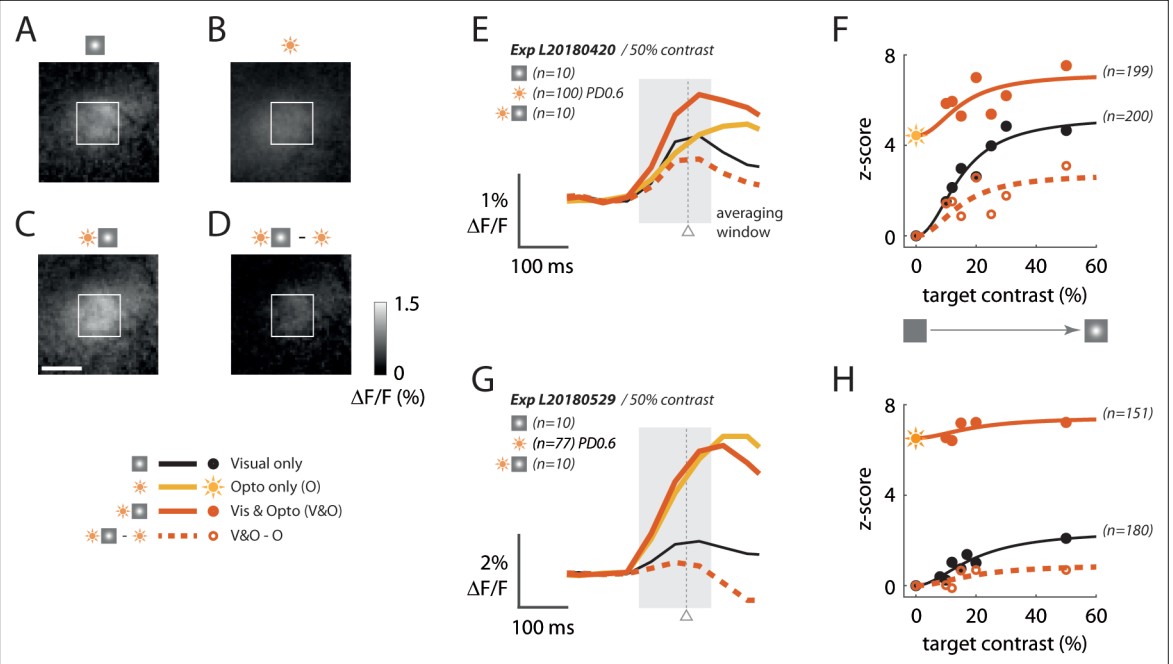

**Figure 5.** Sublinear neural interactions between visual and optogenetic stimulation in macaque V1 consistent with the behavioral masking effect. (**A–D**) GCaMP imaging response maps from an example experiment in Monkey L's C1V1 site to (**A**) the visual target alone at 50% Weber contrast, (**B**) optostim alone at power density (PD) of 0.6 mW/mm², and (**C**) simultaneous visual and optostim. (**D**) Visual-evoked response in the presence of optostim (the simultaneous response in (**C**) with the optostim baseline response (**B**) subtracted). The white rectangle marks the 1 × 1 mm² region of interest (ROI) of cortex selected by maximizing the encompassed visual-only response. The white scale bar marks 1 mm. For (**A–D**), response was averaged over the interval 50–200 ms (shaded area in (**E**)). (**E**) Time course of the GCaMP response averaged within the 1 × 1 mm² window as marked in (**A–D**). The gray triangle indicates the median (saccade) reaction time across all optostim and visual-only trials in this experiment. (**F**) Contrast response functions with and without optostim. Each data point is the time-averaged response expressed as a z-score obtained by normalizing with the standard deviation of the blank (no visual and no optostim) trials. Traces indicate the Naka-Rushton curves fitted to the data (see Materials and methods for details). Sun symbol represents response to optostim when target contrast is zero – that is, response to optostim alone. (**G–H**) are the same as (**E–F**) for a second experiment from Monkey L taken about 6 weeks after the experiment in (**A–D**). In this experiment, the optostim-evoked response was much larger than the visually evoked response. See *Figure 2—figure supplement 2* for time course of $R_{opto}/R_{vis50}$ as a function of the day from C1V1 injection in both monkeys.

In addition to eliciting a large increase in the mean neural response, optostimulation could also cause changes in trial-to-trial response variability. Analysis of the effect of optostim on the variability of the GCaMP-evoked responses shows that optostim tends to increase response variability, but this effect is not present in all experiments (*Figure 6—figure supplement 1*).

Finally, to test for the physiological specificity of the nonlinear interaction, we imaged V1 responses at the control GCaMP-only site (*Figure 6E–H*). We find little or no effect of optostim on the visual-evoked response at the GCaMP-only site even though we could still measure a weak optostim-evoked response (average $R_{opto}/R_{vis5o}$=0.33 at this site). This weak optostim response could reflect a mixture of direct optogenetic activation of horizontal axons originating from the C1V1 site and reaching the GCaMP-only site and/or spread of neural responses elicited by optogenetic stimulation at the C1V1 site through horizontal connections. Future studies with more focal activation of either site would allow one to estimate the contribution of these two possible sources. Overall, the observed physiological specificity is consistent with the specificity of the behavioral effect (*Figure 4*).

## Discussion

Here, we used our novel optical-genetic toolkit for all-optical bi-directional probing of macaque cortex to study simultaneously, for the first time, the neural and perceptual masking effects of optogenetic stimulation in macaque V1. We used a viral-based method to stably co-express in excitatory neurons the calcium indicator GCaMP6f, for optically 'reading' neural population responses at one wavelength, and the red-shifted opsin C1V1, for simultaneously optically 'writing' neural responses

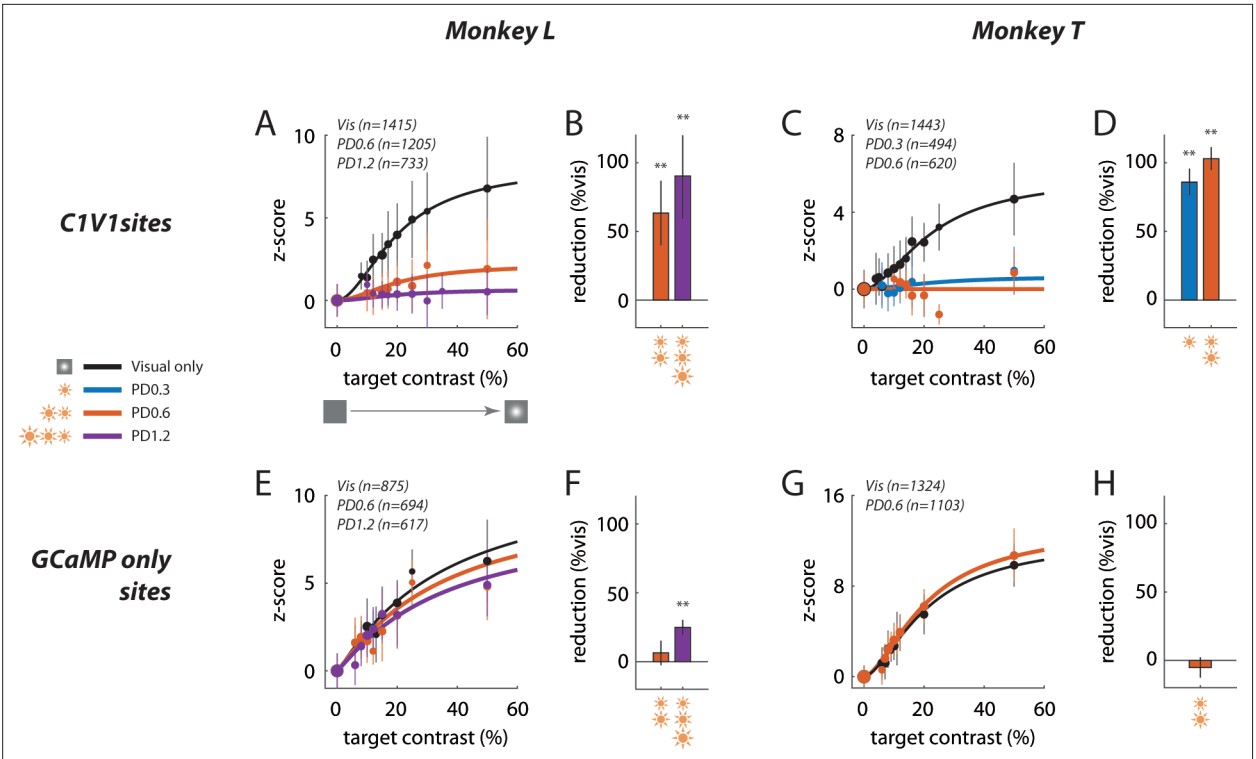

**Figure 6.** Summary of sublinear interaction between visual and optogenetic stimulation in macaque V1. (**A**) Contrast response functions for the visual target alone and for the visual target in the presence of optostim (i.e., visual and optostim minus optostim alone) for Monkey L at the C1V1 site. Response in z-score (as in *Figure 5F and H*) was pooled across all visual-only experiments and across all optostim experiments for each power density (PD) level (in mW/mm$^2$), matching the plotted data in *Figure 3*. The total number of trials pooled for each group is indicated in the legend, and the size of the plotted markers represents the relative number of trials at each target contrast. Error bars represent the standard deviation across trials. Smooth curves indicate the Naka-Rushton function fitted to the data (see Materials and methods for details). (**B**) Percent reduction in overall contrast response with optostim relative to the overall contrast response without optostim (visual stimulation alone), at Monkey L's C1V1 site (see Materials and methods for details). Error bars plot the bootstrapped standard deviation (number of bootstrap runs = 1000). (**C–D**) Same as (**A–B**) for Monkey T's C1V1 site. (**E–H**) are the same as (**A–D**) for respective monkeys at the GCaMP-only sites. Asterisks marks statistical significance of signal reductions using bootstrap analysis. Subtracted optostim baseline in z-score units are (mean ± s.d.): (**A**) 17.4 ± 9.1 [PD0.6] and 37.4 ± 23.4 [PD1.2]; (**C**) 9.9 ± 3.2 [PD0.3] and 12.3 ± 2.1 [PD0.6]; (**E**) 1.9 ± 1.2 [PD0.6] and 3.6 ± 1.2 [PD1.2]; and (**G**) 1.7 ± 1.4 [PD0.3].

The online version of this article includes the following figure supplement(s) for figure 6:

**Figure supplement 1.** Effect of optostim on neural variability.

**Figure supplement 2.** Comparison of evoked GCaMP responses to pulsed and continuous optostim.

at a different wavelength. We then developed a single-photon (widefield) optical system capable of simultaneously reading and writing neural population responses while monkeys perform a demanding visual detection task.

We used our toolkit to test the hypothesis that low-power (peak power density <1 mW/mm$^2$) optostim of the visual cortex can recapitulate the perceptual masking effect of a visual mask on visual detection (i.e., Weber's law), and that this perceptual masking effect is due to simultaneously measured sublinear interaction between visual- and optogenetic-evoked neural responses in behaving macaque V1 (similar to the nonlinearity between two superimposed visual stimuli).

We found that monkeys' detection thresholds are significantly elevated when the visual target is presented simultaneously with low-power optostim. Concurrent optostim and GCaMP imaging revealed that the decline in behavioral detection sensitivity could be attributed to sublinear interaction between the optogenetically and visually driven neural responses in V1; that is, responses evoked by the visual target decrease significantly when riding on top of an optostim-driven response mask. Finally, we find that these neural and behavioral masking effects are spatially selective. The effects are maximal when the target is at the visual field location corresponding to the receptive field of the stimulated neurons, and become weak when the stimulus is moved to a nearby location.

Our experimental approach builds on our ability to maintain, for extended periods (typically >2 years), cranial windows with direct optical and physical access to the cortex of awake, behaving macaques. This approach has multiple advantages. For example, it allows us to perform viral injections under visual guidance and to monitor in real-time the viral spread (*Seidemann et al., 2016*), to use imaging to confirm the GCaMP and opsin expression in parallel and across multiple injections sites, to monitor the long-term stability of the expression, and to calibrate the stimulation parameters based on the optically measured neural responses. One weakness of the current opsin construct is that the eYFP tag that is attached to C1V1 creates a fluorescent baseline that reduces the functional GCaMP signal (*Figure 2C*, *Figure 2—figure supplement 1C*). Future use of an opsin with a red static fluorescent tag will address this issue.

## Perceptual consequences of optogenetic stimulation in V1

The goal of the current study was to examine the behavioral and neurophysiological interactions between visual and optogenetic stimulation in V1. A key feature of our experiment was that the animals were rewarded to report only the visual stimulus, and were thus incentivized to ignore the optostim signal. In contrast, in most previous optostim behavioral experiments, NHPs were rewarded for reporting the optostim signal itself (e.g., *Ju et al., 2018*; *May et al., 2014*). Such studies do not directly test for perceptual substitution by optostimulation. In contrast, in the current study, if the neural representations of the visual stimulus and the optogenetic-evoked response were distinct and separable perceptually, we would have expected the monkeys to learn to effectively ignore the optostim and perform the task with similar accuracy in blocks with and without optostim, as we indeed observed when the target was presented at the nearby retinotopic location corresponding to the GCaMP-only site. Our finding that monkeys show large perceptual masking effects of optostim when the visual target is presented at the receptive field of the optogenetically stimulated neurons, indicates that the optostim-evoked response is largely inseparable perceptually from the visual-evoked response, and that optostim can substitute for a visual mask even when the animal is motivated to ignore it. Thus, our results reveal that optogenetic stimulation of V1 is altering the monkeys' perception of the visual target in a way that is consistent with visual masking. Similar to visual masking, this effect is spatially selective, and the monkeys do not seem to be able to learn to compensate for the presence of the optogenetic (*Figure 2—figure supplement 2*).

Importantly, we were not attempting to generate an optogenetic-evoked response that is indistinguishable from the response to the visual target. Because the target size (0.33° FWHM) was smaller than the average receptive field of V1 neurons at these eccentricities (0.58° FWHM; *Chen et al., 2012*), the cortical area responding to the target was approximately equal to the cortical point image, and had an area at half max of ~4 mm$^2$ (*Palmer et al., 2012*), which is about twice the size of the area at half max of the optostimulation-evoked responses (*Figure 2C*; *Figure 2—figure supplement 1C*). Therefore, it is likely that optostim evoked a response in a cortical area that is smaller than the response evoked by the visual target, and that even within the overlapping region the two stimuli elicited responses with different spatial profiles. In this respect, optostim masking is similar to visual masking, where the mask does not have to be identical to the target to lead to a perceptual masking effect.

## Relevance for sensory cortical neuroprostheses

Our results have important implications for the development of future optical brain-computer interfaces (oBCI; *O'Shea et al., 2017*). First, while previous results suggest that virally delivered genes can lead to stable foreign protein expression for extended periods in humans and NHPs, we document here stable co-expression of reporters and actuators for periods that exceed the lifespan of our chambers (typically >2 years; Monkey T's chamber is still active more than 30 months from chamber opening with continued stable expression of GCaMP and C1V1; *Figure 2—figure supplement 1*), and possibly for the lifespan of the animal.

Second, effective optostim at low-power densities is important for stable, long-term use in future implantable oBCIs. Behavioral and physiological effects of optogenetics in macaques typically require high stimulation light power densities (e.g., *Ruiz et al., 2013*; *Nassi et al., 2015*; *Jazayeri et al., 2012*; though see *Ju et al., 2018* for a demonstration of a behavioral effect at low power), leading to concerns about possible tissue damage (*Podgorski and Ranganathan, 2016*) and reduced

effectiveness of optostim over time. High light level may also limit the use of future implantable devices due to power requirements. We demonstrate that very low optical power levels are sufficient to elicit reliable neural responses within the biologically relevant dynamic range of neural responses to visual stimuli and to replace a visual mask with optostim in a visual detection task. In fact, the optostim power densities reported here overestimate the actual effective power densities used in our study. The optostim light was flashed with a duty cycle of 22%; therefore, the time average of the power density was only a fraction of the maximum power density reported. Preliminary results suggest that the optostim-evoked neural responses are proportional to the time average of the power density of the stimulation light and are relatively insensitive to the temporal parameters of the stimulation (*Figure 6—figure supplement 2*). In addition, we used orange light at 580 nm to stimulate C1V1 in order to prevent the optostim light from contaminating our simultaneous calcium imaging. This orange light is shifted from the peak excitation of C1V1 (~530 nm) and this further reduced the effective power of the optostim.

## Nonlinear interaction between visual and direct optostim response in macaque V1

Our results reveal clear sublinear interaction between GCaMP-measured neural responses elicited by visual stimuli and by direct optogenetic stimulation in V1. We previously showed that widefield GCaMP signals are approximately linearly related to the locally pooled spiking activity in macaque V1 (*Seidemann et al., 2016*). Therefore, the observed sublinear interaction is likely to reflect a sublinear interaction at the level of V1 spiking activity. Consistent with this possibility, pilot electrophysiological results demonstrate strong sublinear interaction between visual- and optogenetic-evoked multi-unit spiking activity at the C1V1 site in Monkey L (data not shown).

While it is well known that neural responses to visual stimuli are highly nonlinear and are consistent with luminance and contrast gain control (e.g., *Albrecht and Hamilton, 1982*; *Purpura et al., 2009*), some of the nonlinearities observed in V1 response are likely to be inherited from its ascending inputs. It is therefore unclear how much of the gain control is implemented in V1 and whether the interactions between visual- and optostim-evoked responses are similar to those evoked by two visual stimuli. Our results are consistent with a significant contribution of V1 circuits to gain control, or normalization. However, we cannot rule out that some of the observed sublinearity could be due to V1 neurons that are driven to their physiological limit by optostim.

The observed sublinear interaction between visual- and optogenetic-evoked GCaMP responses in V1 is consistent with recent findings from a study by Nassi et al., where they used electrophysiology to study the interactions between visual- and optogenetics-evoked responses in fixating macaque V1 (*Nassi et al., 2015*). However, in that study, the light power densities required to elicit optogenetic-evoked responses were one to two orders of magnitude higher than in the current study. Such high power densities raise concerns regarding tissue heating, which could affect the nature of the interactions between visual- and optogenetic-evoked responses in V1. Therefore, our results, in addition to demonstrating a novel perceptual interaction between visual and optogenetic stimulation in V1, significantly extend the results of Nassi et al., and provide an important confirmation of their main finding. Multiple factors could contribute to the different sensitivity to optostim observed in the two studies, including differences in opsin expression levels, differences in the size of the neural population stimulated and/or monitored in the two studies, and difference in the visual stimulus (drifting gratings vs. a small Gaussian target in the current study).

Our results contrast with those of a recent study which used optostim and electrophysiology to examine the perceptual and neural effect of V1 optostim in monkeys performing detection of very dim visual gratings on a dark background (more than two orders of magnitude darker than in the current study; *Andrei et al., 2019*). In that study, optostim caused a small *improvement* in detection when stimulus orientation was near the preferred orientation of the optostimulated neurons, but had no effect when it was far. Electrophysiological measurements revealed no interaction between average visual- and optogenetic-evoked responses, but instead, optostimulation induced a reduction in noise correlations between pairs of neurons only when stimulus orientation was near the preferred orientation of the optostimulated neurons. These contrasting results are not surprising given that the two studies examined the interactions between visual and optogenetic stimulation under very different regimes of cortical activity.

## Low-power optostim-evoked responses exceed responses to the visual stimulus

We find that even though optostim engages gain control mechanisms in V1, low-power optostim can elicit GCaMP population responses that far exceed the responses to our Gaussian targets (*Figure 5G and H*; *Figure 2—figure supplement 2B-D*). This is an interesting observation, but it does not necessarily imply that optostim can drive V1 population responses to a range that exceeds the population response to any visual stimulus. First, even for the small Gaussian stimuli used in the current study, it is likely that population responses can increase significantly by increasing the target amplitude while lowering the background luminance. Because the background luminance in the current study was high relative to the target amplitude, the response evoked by our Gaussian targets was moderate.

Second, it is possible that other classes of visual stimuli can elicit population responses that are higher than the response to a Gaussian target. However, we recently found that localized low spatial frequency stimuli are surprisingly effective at driving V1 population activity (*Benvenuti et al., 2018*). Widefield calcium imaging signals at each cortical location reflect the weighted sum of neural activity pooled over a Gaussian-shaped region with a space constant of ~0.2 mm (*Seidemann et al., 2016*; *Chen et al., 2012*). Therefore, the GCaMP-evoked neural response reflects the pooled spiking activity of a large population of neurons with diverse tuning properties. The Gaussian stimuli used here are suboptimal for individual V1 neurons because they are unoriented. Even though they drive individual neurons weakly, they can lead to robust population response in V1 because they activate a large fraction of the neurons at the corresponding retinotopic location. In contrast, visual stimuli that are optimized for a subset of V1 neurons, such as oriented Gabor patches, activate strongly a small subpopulation but activate poorly most other neurons. Therefore, Gaussian targets could be surprisingly effective at driving locally pooled V1 population responses. Consistent with this possibility, detection thresholds for Gaussians are comparable to detection thresholds for Gabor patches (*Watson and Ahumada, 2005*).

The relative range of population responses achievable by direct optogenetic stimulation of excitatory V1 neurons vs. visual stimuli is an important question for future research. If future results reveal that optostim of excitatory neurons can drive V1 population responses outside of the range achievable with visual stimuli, such a result could reflect several causes. One possibility is that optostim does not engage gain control to the same extent as visual stimuli because during visual responses V1 receives ascending inputs that have already undergone through gain control mechanisms in the retina and LGN. A second possibility is that optogenetic stimulation can partially override gain control mechanisms that operate within V1.

## Larger optostim masking effect on V1 neural detection sensitivity than on behavioral sensitivity

Our results reveal a larger masking effect of optostim on V1 neural sensitivity than on behavioral sensitivity. In both monkeys, low-power optostim can effectively erase the visual-evoked responses from our GCaMP measurements (i.e., reduce target-evoked responses by 100%; *Figure 6B and D*), yet both monkeys could still perform the detection task with optostim (albeit with elevated thresholds). These results suggest that our widefield GCaMP imaging is not capturing all of the visual signals that are available for detection in V1. This could reflect visual responses that spread laterally over a larger region than the area affected by optostim and/or target-evoked signals that are deeper in the cortex and are therefore inaccessible to widefield imaging. In addition, it is possible that even within the affected region, some target-related signals that are present at the single neuron level are eliminated by the indiscriminate pooling of widefield imaging over large populations of neurons.

## Future directions

While our study and previous findings (*Nassi et al., 2015*) reveal clear nonlinear interaction between visual- and optostim-evoked responses in V1, additional research is needed in order to achieve a more complete understanding of these nonlinearities and their underlying mechanisms. Our study demonstrates the importance of combining optogenetic stimulation and optical imaging to calibrate the stimulation so that the evoked neural response falls in a biological-relevant regime. Future studies could take further advantage of this simultaneous read-write capability and use real-time close-loop feedback to optimize the stimulation pattern. In the current study, we used spatially homogeneous

optogenetic light stimulation to generate a localized mask of optostim-evoked neural responses. An important future direction is to replace the current source of the optogenetic light stimulation with a projector that would allow one to generate arbitrary spatiotemporal patterns of light stimulation (*Huang et al., 2014*; *Chen et al., 2019*), thus furthering the ability to mimic with optostim the pattern of population responses elicited by visual stimuli at relevant topographic scales such as retinotopy and orientation columns. This form of patterned optostim could be used in the future to causally test specific hypotheses regarding the role of topography in sensory cortical representations.

## Materials and methods

All procedures have been approved by the University of Texas Institutional Animal Care and Use Committee and conform to NIH standards.

### Viral injection

Two macaque monkeys (Monkey L and Monkey T; *Macaca mulatta*, male, 7–8 years of age during experimentation, weighing 11 and 7 kg, respectively) were implanted with a custom recording chamber placed over cranial windows for direct physical and optical access to the awake macaque primary visual cortex (V1; *Figure 2A*; *Figure 2—figure supplement 1B*). Our general methods for viral injections have been described previously (*Seidemann et al., 2016*). For neural reading, each monkey was first injected with ~2 µl of a GCaMP6f (*Chen et al., 2013*) viral vector per injection site (Monkey L with AAV1-CaMKIIa-NES-GCaMP6f [Deisseroth lab], and Monkey T with AAV1-CaMKIIa-GCaMP6f [Zemelman lab]). Our previous work revealed that this virus leads to GCaMP expression in ~90% of excitatory neurons in macaque V1 (*Seidemann et al., 2016*). Progress of the GCaMP6f expression was monitored by measuring visual response to large gratings flashing at 4 Hz (*Figure 2C*; *Figure 2—figure supplement 1C*). Initial expression of GCaMP6f was observed at around 6–10 weeks post injection. Stable functional response levels were reached 4–8 weeks after (*Figure 2D*; *Figure 2—figure supplement 1A*) and lasted for the lifespan of the chamber (>2 years in both monkeys).

For neural writing, following stable GCaMP6f expression, a second viral injection of ~2 µl was performed collocated to one of the initial GCaMP6f expression sites. This viral vector contained a red-shifted ChR-based opsin C1V1 (*Packer et al., 2012*) (AAV5-CaMKIIa-C1V1(t/t)-TS-eYFP [Deisseroth lab] for both monkeys). Progress of the C1V1 expression was monitored by measuring the vector's eYFP static florescence and evoked response of the collocated GCaMP6f expression to direct optical stimulation. Initial C1V1 expression was observed 4–6 weeks post injection, and again took 4–8 weeks for stable functional response levels to be obtained (*Figure 2—figure supplement 1*). Because the eYFP emission spectrum overlaps with that of GCaMP, the baseline fluorescence ($F$) at the co-expression site was significantly elevated after the C1V1 expression, leading to an apparent reduction in the amplitude of the visual-evoked responses at this site relative to the GCaMP-only site (*Figure 1C*, top). However, the change in fluorescence in response to the visual stimulus ($\Delta F$) remained comparable at the two sites (*Figure 1C*, middle), indicating stable expression of both the GCaMP and C1V1 (see also *Figure 2—figure supplement 1C*).

### Widefield GCaMP imaging and optical stimulation

The experimental technique for widefield (one-photon) fluorescent optical imaging of neural response in awake-behaving macaques was adapted from previous studies (*Seidemann et al., 2016*; *Seidemann et al., 2002*). Briefly, each monkey was implanted with a metal head post and a metal recording chamber located over the dorsal portion of V1 (*Figure 2A*; *Figure 2—figure supplement 1B*), a region representing the lower contralateral visual field at eccentricities of 2–5°. Craniotomy and durotomy were performed. A transparent artificial dura made of silicone was used to protect the brain while allowing optical access for imaging (*Arieli et al., 2002*).

Widefield optical imaging employed a double-SLR-lens-macro system with housing for dichroic mirrors in between the two SLR lenses (*Figure 2B*). The combination of a 50 mm fixed-focus objective lens (cortex end, Nikkor 50 mm f/1.2) and an 85 mm fixed-focus objective lens (Canon EF 85 mm f/1.2 L USM) provided 1.7× magnification, corresponding to imaging approximately an $8 \times 8$ mm$^2$ area of the cortex (using PCO Edge 4.2 CLHS sCMOS camera).

An appropriate set of interference filters and dichroic mirrors were employed to adapt the system for GCaMP6f imaging. The illumination (excitation) light was filtered at 480 nm (ET480/20×), through a long-pass dichroic mirror at 505 nm (505LP), and GCaMP6f emission was filtered at 520 nm (ET520/20m).

To further adapt the apparatus for simultaneous neural reading and writing, a second dichroic box was inserted on the camera side to accommodate the optical stimulation light path. This light path was set up with a filter set for C1V1 excitation at 580 nm (ET580/20×), through a short-pass dichroic mirror at 550 nm (550SP). Additionally, the eYFP static florescence co-expressed with the C1V1 was tracked with a 505 nm excitation filter (ET505/20×), a 455/520/600 nm multiband dichroic (69-008BS) and a 540 nm emission filter (ET540/30m). All filter parts were obtained from Chroma Technology.

Illumination for GCaMP read-out was supplied with a broadband LED light source (X-Cite120LED). Optical stimulation was also applied with a broad-spectrum LED light source (X-Cite120LED for Monkey L, and X-Cite Xylis for Monkey T). Due to overlap between GCaMP6f and C1V1 absorption spectra, it was observed that excessive GCaMP read-out blue illumination can lead to decreased visual target detection rate (*Figure 2—figure supplement 3*). Therefore, imaging was conducted with minimal GCaMP illumination power (0.05 mW/mm² in Monkey L and 0.01 mW/mm² in Monkey T) at a low camera saturation level (1–5%) and a low 20 Hz frame rate. Note that the peak power density of the blue light was an order of magnitude lower than the lowest level of orange light used in each animal. Combining this with the lower effectiveness of blue light relative to orange light in activating the red-shifted opsin C1V1, the effect of the blue light was likely to have been negligible. However, because our experiment focuses on the comparison between visual-only and visual-plus-optostim trials, and the blue light was present in both trial types, even if it had a small perceptual effect, this has no impact on the conclusions of our study.

Imaging was conducted using software developed in-house running in Matlab R2018b (utilizing Image Acquisition Toolbox). Data acquisition was time locked to the animal's heartbeat (EKG QR up-stroke). Raw images were captured at 2048 × 2048 resolution, binned to 512 × 512. EKG was measured using HP Patient Monitor (HP78352C).

## Behavioral task with optical stimulation

The monkeys detected a white Gaussian target on a uniform gray background (*Figure 2C*). The monkeys' performance in blocks of trials with no optostim (*Figure 1C*, top right) was compared with their performance in blocks of trials in which an optostim 'mask' appeared on all trials (both target-present and -absent trials). Note that even in the optostim blocks, the monkeys were rewarded only for reporting the presence or absence of the visual target (*Figure 1C*, bottom right).

The Gaussian target (0.33° FWHM) was smaller than the average V1 receptive field size (which is similar to the population receptive field) at these eccentricities (~0.58° FWHM; *Chen et al., 2012*). The target was positioned at corresponding retinotopic coordinates of the C1V1 expression site, which was 1.5° eccentricity (–35° from the right-hand horizontal meridian) in Monkey L and was 2.7° eccentricity (–46° from the right-hand horizontal meridian) in Monkey T. For the GCaMP-only experiments, the visuals stimulus was centered at the retinotopic location corresponding to a nearby cortical site with only GCaMP6f expression (no C1V1 injection/expression). This was 2.0° eccentricity (–45° from the horizontal meridian) for Monkey L, and 2.0° eccentricity (–40° from the horizontal meridian) for Monkey T. The distances between the centers of the C1V1 and GCaMP-only sites were 3 mm in Monkey L (corresponding to 0.6° distance between receptive field centers) and 4 mm in Monkey T (corresponding to 0.7° distance between receptive field centers). While the Gaussian targets at the C1V1 and the GCaMP-only sites in each monkey did not overlap in the visual field, the population receptive fields at these two locations partially overlapped, so that the visual target at one location partially activated some of the neurons at the other location.

Each trial began with fixation on a bright 0.1° square. An auditory tone and the dimming of the fixation square cued the monkey to the start of the detection task trial; 250 ms later, the Gaussian target was presented on 50% of the trials. The monkeys were trained to stay at the fixation cue on target-absent trials or make a saccade to, and hold gaze (for 150 ms) at, the target position to indicate target detection (with a 75 ms minimum allowed reaction time). When the target was present, it remained on screen for a maximum of 250 ms or was extinguished upon the monkeys' saccade (*Figure 1C*, right). The monkey was given 600 ms to make the saccade or to hold fixation, and was subsequently

rewarded on correct choices (stay [correct reject] on target-absent trials, and saccade to target [hit] on target-present trials). The size of the fixation window ($X \times Y$) was 1.8–2.4° × 1.8–2.8° and the saccade target window was 2.4–3.0° × 2.4–4.5°.

The strength of the Gaussian target is reported in Weber contrast:

$$c = \frac{L_{max} - L_{background}}{L_{background}}$$

The contrast of the Gaussian target was varied to measure the psychometric curve. Trials were presented in blocks, with the target contrast held constant within each block. Blocks were pooled and the detection threshold was estimated from the monkey's choices by maximum likelihood fitting of the following equations for hit rate $P(Hits)$ and correct rejection rate $P(CRs)$:

$$P\left(HITs\right) = \Phi\left(\frac{1}{2}\left(\frac{c}{\alpha}\right)^{\beta} - \delta\right), P\left(CRs\right) = \Phi\left(\frac{1}{2}\left(\frac{c}{\alpha}\right)^{\beta} + \delta\right),$$

where $\Phi$ is the standard cumulative normal function, $\alpha$ the detection threshold defined at $d$-prime ($d'$) = 1, $\beta$ the steepness of the psychometric curve, and $\delta$ the monkey's bias from the optimal criterion (in units of $d'$). If either the proportion of hits or correct rejections has a value of 0 or a value of 1.0, then the likelihood becomes undefined. To avoid leaving out the data in these conditions, we scaled all the proportions to be between 0.005 and 0.995 ($P(x) = 0.005 + 0.99 \cdot P(x)$). The optimal criterion is defined when $P(HITs)$ and $P(CRs)$ are equal for all target contrasts ($\delta = 0$).

The monkeys' baseline detection threshold was assessed in blocks of trials with no optostim. In separate blocks on the same day, the detection threshold with optostim was obtained for comparison.

Psychometric curves from visual-only and optostim trials were fitted jointly, allowing different thresholds ($\alpha$) and biases ($\delta$), but sharing the same steepness parameter ($\beta$). (Separate steepness parameters had overlapping confidence intervals.) For visualization, the fitted $d'$ values were converted back to bias-corrected %correct values by setting $\delta = 0$ in the fitted cumulative normal function for $P(HITs)$ and $P(CRs)$ (*Figures 3C, H, 4C and H*). The change in the behavioral threshold in visual-plus-optostim trials was characterized as the fraction increase above the visual-only threshold (normalized threshold, *Figures 3D, I, 4D and I*):

$$\rho_{behavior} = \frac{\alpha_{optovis}}{\alpha_{vis}}$$

Bootstrap resampling of trials was used to estimate the standard error of the fitted parameters (more details in Statistics subsection). For normalized thresholds, the mean thresholds and standard error were normalized with respect to the mean visual-only threshold (fixed value for each plot); consequently, the error bars around the normalized visual-only threshold reflect the bootstrapped variability of the visual-only threshold.

The visual target was presented 250 ms after the temporal cue, whereas the optostim was applied 290 ms after the cue. This 40 ms delay in optostim insured that optostim-evoked V1 response coincides approximately with the onset of the visual-target-evoked response on target-present trials (*Figure 5E and G*). Optostim was presented in pulses of 5 ms spaced every 22.5 ms (~44 Hz), and was extinguished after 210 ms (coinciding with maximum visual target duration) or on the onset of saccade (coinciding with extinguishing the visual target). Pilot studies, in which we compared widefield GCaMP response to pulsed and continuous optostim, suggest that the V1 response is approximately proportional to the time average of the optostim power density (*Figure 6—figure supplement 2*). These pilot studies suggest that the effective power density was about 25% of the peak power density reported here.

The power density of optostim was varied as an experiment parameter. Power density (in mW/mm²) was measured through all the optical filters/apparatus at the plane of the imaged cortex, with the total power read from a light meter (ThorLabs PM100D with S170C sensor) and divided by the illuminated area (circular disk: diameter 13 mm for C1V1 optostim light, and 19 mm for GCaMP light).

Experiments were conducted with custom code using TEMPO real-time control system (Reflective Computing). The visual stimulus was presented on a Sony CRT (1024 × 768 @ 100 Hz), distanced 108 cm from the animal (50 pixels-per-degree), with mean luminance 50 cd/m². The visual stimulus was generated using in-house real-time graphics software (glib). Eye tracking was implemented with an EyeLink 1000 Plus.

## GCaMP neural response analysis

For each trial, an image sequence was captured for a total of 1.2 s including pre-stimulus and post-stimulus frames. The image sequence was analyzed to extract the visual and optical-stimulation-evoked response using a variant of the previous reported routines (*Seidemann et al., 2016*; *Chen et al., 2012*; *Chen et al., 2006*). Additional stages were added to the routine to reduce known sources of noise.

The preprocessing stage of image analysis involved the following steps: image stabilization, down-sampling, $\Delta F/F$ normalization, heartbeat removal, and pre-stimulus anchoring. Image stabilization is a new routine we use to de-accentuate blood vessel edges in the $\Delta F/F$ response map caused by micro movements of the camera and/or the cortex during imaging.

The image intensity across time at each individual pixel was modelled with separable motion-free ($I_{xy0}[t]$) and motion-related ($\vec{\alpha}_{xy}.\vec{v}[t]$) components as follows:

$$I_{xy}[t] = I_{xy0}[t] + \vec{\alpha}_{xy}.\vec{v}[t]$$

For each trial, a single global motion vector $\vec{v}[t]$ was obtained by estimating the translational motion of the center portion of the images (1/4 of the imaging area). The motion coefficients $\vec{\alpha}_{xy}$ for each pixel was then obtained using least squares fitting to the model. The motion-corrected image is $I_{xy0}[t]$. This approach to image stabilization (compared to traditional image registration approach) has the advantage of correcting for non-rigid movements (rotations, expansion/contractions, affine transformation, local distortions, etc.) and sub-pixel motion.

Heartbeat removal employed a similar approach. The heartbeat artifact is much larger for imaging in the GCaMP spectrum than for imaging in the voltage-sensitive dye spectrum. Due to slight differences in heartrate between trials, despite synchronizing imaging start on a QRS upstroke of the EKG, subsequent heartbeats fall on different frames across trials, which create trial-to-trial variations that cannot be removed with a simple blank-subtraction approach. Nonetheless, the EKG synchronized start of trials acted as a common pivot point across trials, and as the heartrate changed, the heartbeat artifact stretched or compressed in time like an accordion from this pivot. Consequently, the pixel intensities at the same frame index across trials varied in a predictable manner with respect to the heartrate in each trial. This was exploited in the following model, putting the heartrate-free stimulus-driven response ($S_{xyt}(s)$, to stimulus *s*) and heartrate-related ($f_{xyt}(HR[k])$) intensity values as additive components:

$$I_{xyt}[k] = S_{xyt}(s) + f_{xyt}(HR[k])$$

where $HR[k]$ is the estimated heartrate on trial *k*. The heartrate-related component (artifact) was modelled with a fourth-order polynomial of the heartrate:

$$f_{xyt}(HR[k]) = a_{xyt}^4 HR^4[k] + b_{xyt}^3 HR^3[k] + c_{xyt}^2 HR^2[k] + d_{xyt} HR[k]$$

Trials from all stimulus conditions were pulled from the same recording session, from which the heartrate-free stimulus-driven responses and polynomial coefficients were estimated simultaneously using least squares fitting; and then the estimated heartrate-related component was subtracted out from the data.

The measured physiological signal was $\Delta F/F$. For each trial, the average florescence over frames 0–150 ms prior to stimulus onset (three frames [F5–F7]) was used as the baseline reference ($F_0$). $\Delta F/F$ was calculated as follows:

$$\frac{\Delta F}{F}(t) = \frac{F(t) - F_0}{F_0}$$

The neural response was averaged over the 1 × 1 mm² cortical area presenting the best visually driven response. The mean response within this spatial window was used to characterize the time course of the neural response (*Figure 5E and G*). Slow drifting responses over time were minimized with a linear de-trending routine using the following model of the stimulus-driven response ($R_S[t]$) for each stimulus condition *S*:

$$R[t] = R_S[t] + (m_S + m_k) \times t$$

This model treats the linear trend as two component gradients: a linear trend on the conditional means ($m_S$) shared across all trials from the same stimulus condition $S$, and additional linear trends in individual trials ($m_k$). $m_k$ was estimated by minimizing the squared errors within trials for the same stimulus conditions, using the data up to the frame that a saccade was made. Pre-stimulus frames were used to estimate $m_S$, which constrained all the conditional means ($R_S[t]$) to a flat, zero pre-stimulus baseline. The detrending routine removed both components of the linear trend in the data.

The average response over the three frames from 50 to 200 ms post-stimulus was used to represent the neural signal pertaining to the animal's behavioral decision. Responses beyond this range were not used as they are generally post-saccade (median saccade times were 149 and 204 ms, respectively, for Monkey L and T; **Figure 3—figure supplement 2**).

For pooling across experiments, the neural response in each trial was converted to a $Z$-score that was calculated with respect to the standard deviation of the blank trials (no visual and no optostim, $\sigma_{blank}$) from the same experiment:

$$Z = \frac{R - R_{blank}}{\sigma_{blank}}$$

The $Z$-scores were then pooled across experiments for further analysis (**Figure 5F and H**; **Figure 6**).

The Naka-Rushton function was used to characterize neural response as a function of visual stimulus contrast ($c$):

$$Z(c) = Z_{max} \frac{c^n}{c^n + c_{50}^n}$$

The contribution of the visual stimulus ($Z_{optosub}$) to simultaneous visual and optostim response ($Z_{optovis}$) was calculated by subtracting the mean optostim-only baseline response ($Z_{optobase}$) from the simultaneous visual and optostim response:

$$Z_{optosub} = Z_{optovis} - Z_{optobase}$$

The reduction of the visually evoked response under simultaneous optostim (**Figure 6**) was expressed as proportional reduction from the visual-only response, calculated by finding the least squared fit across target contrast:

$$\rho_{physiology} = \frac{Z_{vis} - Z_{optosub}}{Z_{vis}}$$

All analyses were done using Matlab R2018a.

## Statistics

Two animals were examined to verify the consistency of experimental approach and results. Multiple recordings were made from the same animals. The number of recordings was based on previous experience; no statistical method was used to predetermine sample size.

**Figure 2**, **Figure 2—figure supplement 1** outline our experiment timeline. Each experiment condition (optostim power level × cortical site) was repeated at least three times (in separate days) in each animal, pairing with a visual-only block each day. The number of trials collected each day varied between 200 and 600 trials (typically 400) depending on the animal's motivation. Trials were excluded when the monkey did not follow the task routine, when EKG signal was noisy, and when excessive motion artifact was detected in the imaged data. The total numbers of trials used are reported in each figure.

Where bootstrapped resampling was applied, the data was resampled 1000 times with replacement within trials of the same stimulus condition (visual target amplitude and optostim level). For **Figures 3 and 4** and **Figure 2—figure supplement 2**, the behavioral detection threshold and criterion were refitted to each bootstrapped resample of the monkey's behavioral decisions. For **Figure 6**, **Figure 2—figure supplement 2**, the GCaMP response reduction was recalculated on each bootstrapped resample of the GCaMP response. The error bars included in the same figures are the standard deviations of the bootstrapped mean (equivalent to standard errors). For statistical significance, the relevant comparison was made for each bootstrap resample (i.e. 1000 comparisons), and the

proportion that agreed with the null hypothesis was used as the p-value for statistical inference, adjusted for multiple comparisons using Bonferroni's method where required. Statistical analyses were conducted in Matlab (R2018a).

## Acknowledgements

We thank Tihomir Cakic, Kelly Todd, and members of Seidemann laboratory for their assistance with this project. We thank Boris Zemelman for the GCaMP6f viral construct used in Monkey T. This work was supported by NIH grants EY-016454 to ES, EY-024662 to WSG and ES, BRAIN U01-NS099720 to ES and WSG, and DARPA-NESD0-N66001-17-C-4012 to ES.

## Additional information

### Funding

| Funder | Grant reference number | Author |
| --- | --- | --- |
| National Eye Institute | EY-016454 | Eyal Seidemann |
| National Eye Institute | EY-024662 | Wilson S Geisler |
| National Institutes of Health | BRAIN U01-NS099720 | Wilson S Geisler Eyal Seidemann |
| DARPA-NESD | N66001-17-C-4012 | Eyal Seidemann |

The funders had no role in study design, data collection and interpretation, or the decision to submit the work for publication.

### Author contributions

Spencer Chin-Yu Chen, Conceptualization, Data curation, Formal analysis, Investigation, Methodology, Software, Validation, Visualization, Writing – original draft, Writing – review and editing; Giacomo Benvenuti, Conceptualization, Data curation, Methodology, Writing – review and editing; Yuzhi Chen, Conceptualization, Data curation, Investigation, Methodology, Resources, Software, Supervision, Writing – review and editing; Satwant Kumar, Data curation, Formal analysis, Investigation, Methodology, Software, Validation, Visualization, Writing – review and editing; Charu Ramakrishnan, Karl Deisseroth, Methodology, Resources, Writing – review and editing; Wilson S Geisler, Conceptualization, Formal analysis, Funding acquisition, Supervision, Visualization, Writing – original draft, Writing – review and editing; Eyal Seidemann, Conceptualization, Funding acquisition, Investigation, Methodology, Project administration, Supervision, Validation, Visualization, Writing – original draft, Writing – review and editing

### Author ORCIDs

Giacomo Benvenuti http://orcid.org/0000-0002-5234-6260
Eyal Seidemann http://orcid.org/0000-0003-2841-5948

### Ethics

All procedures have been approved by the University of Texas Institutional Animal Care and Use Committee (IACUC protocol #AUP-2016-00274) and conform to NIH standards.

### Decision letter and Author response

Decision letter https://doi.org/10.7554/eLife.68393.sa1
Author response https://doi.org/10.7554/eLife.68393.sa2

## Additional files

### Supplementary files
• Transparent reporting form

## Data availability

The data and Matlab code for visualization are available on Dryad Digital Repository, https://doi.org/10.5061/dryad.00000003h.

The following dataset was generated:

| Author(s) | Year | Dataset title | Dataset URL | Database and Identifier |
|---|---|---|---|---|
| Chen S, Benvenuti G, Chen Y, Kumar S, Ramakrishnan C, Deisseroth K, Geisler W, Seidemann E | 2022 | Data from: Similar neural and perceptual masking effects of low-power optogenetic stimulation in primate V1 | https://doi.org/10.5061/dryad.00000003h | Dryad Digital Repository, 10.5061/dryad.00000003h |

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
