## [Editor Report]

This work examined a long-standing technical and conceptual question in systems neuroscience: can artificial perturbation of primary sensory cortex (in this case V1) mimic the perceptual effects of natural sensory stimulation? This technically impressive work combined optogenetics and visual psychophysics in monkeys to show that certain controlled patterns of V1 simulation can recapitulate a relatively simple visual perceptual effect involving visual masking. The results provide a proof-of-concept for a new set of approaches for studying the neural basis of visual perception.

---

## [Decision Letter]

**Decision letter after peer review:**

Thank you for sending your article entitled "Similar neural and perceptual masking effects of low-power optogenetic stimulation in primate V1" for peer review at *eLife*. Your article is being evaluated by 2 peer reviewers, and the evaluation is being overseen by a Reviewing Editor and Joshua Gold as the Senior Editor. The reviewers have opted to remain anonymous.

Essential revisions:

1. There were concerns about your interpretation of your results in terms of "a sublinear interaction between visual and optogenetic evoked V1 responses" without knowing/considering the relationship between the calcium signals you measured and the underlying spiking activity. You will see that one reviewer requests additional imaging experiments with simultaneous electrophysiological recordings to measure a) the linearity of the relationship between the calcium signal and spiking activity and b) what fraction of cells show network-mediated suppression (as opposed to expected excitation). Upon discussion, the reviewers agreed that such measures would be helpful but are not necessarily crucial. Instead, there may already be information about this particular calcium indicator in the literature, and it was felt that only a small fraction of neurons were likely to be inhibited by the excitatory opsins. It is requested that you consider these factors (particularly the calcium/spiking relationship) more explicitly in your analyses.

2. Show the results of the neural and behavioral data for the power titration experiments.

3. Present the behavioral data with and without the bias-correction and include a figure with the false-alarm rates.

4. Include details about the receptive field properties of the stimulated neurons.

5. Expand the descriptions of the various methodological details (virus injections, behavioral bias-correction methods, etc.).

6.Consider the possibility that the behavioral impairment involves higher-level processing (e.g., where an optogenetically induced percept interferes with visual processing).

7. Consider addressing more directly the nature of the visual percept produced by the optogenetic stimulation. For example, it was suggested that it would be useful to compare the optogenetically induced mask with a purely visual mask. We are not requiring such additional experiments but raise them as a possibility to consider, because it was thought that they would be extremely helpful in providing an approximate measure for the brightness of the optogenetically induced visual percept.

Please see below for more details.

*Reviewer #1:*

The monkeys were trained to detect a target with or without optostimulation of neurons of which the receptive field overlapped the target. The optostimulation impaired target detection and strongly reduced the Ca population response to the visual target (assessed by the difference between the response to the combined opto- and visual stimulation and optostimulation-only). They interprete the reduced visual response with optostimulation in terms of a nonlinear masking effect. Basically, it is the same effect as described previously in terms of divisive normalization in a combined recording- optostimulation study (Nassi et al., Neuron, 2015). However, several issues make the interpretation of the current data difficult. First, it is unclear to me to what extent the relationship between the spiking responses and the measured Ca signal is linear. If this relationship is nonlinear, then it is not straightforward to assess the non/sub-linearity of the spiking activity using the Ca signal. Second, previous optogenetic studies in visual cortex showed that some neurons are excited by optostimulation while others show inhibition, likely resulting from inhibition by interneurons receiving excitatory input from the optostimulated pyramidal neurons. The population Ca signal in the present study may thus reflect a quite heterogeneous set of different responses to the optostimulation and visual stimulus, which makes an assessment of the response of neurons to the different stimulation conditions and linking it with behavior not trivial. Both these concerns can be addressed by having single unit ephys recordings in addition to the Ca population signal. Third, the title of the paper suggests a similar behavioral and neural effect of optogenetic stimulation. Indeed, both read-outs show an impairment of visual responses with optogenetic stimulation, but the similarity ends when making a quantitative comparison of the two read-outs since the behavioral effect is weaker than the neural one. The authors discuss the possible reasons of this incongruency of the behavioral and neural effects. Again, to sort this out laminar ephys would help, e.g. to see whether neurons not affected by the optostimulation, e.g deeper than those generating the Ca signal, could support the behavior.

In sum, this work presents a technical advance in nonhuman primate neurophysiology but its conceptual advance is limited.

1. More information about the data analysis of the behavioral results should be provided. The authors employ a go- nogo task which is known to suffer from response biases, which can differ between animals. The authors present bias-corrected % correct values in Figures 3 & 4 but do not explain how these were computed. In fact, I find it difficult to understand how monkey L can have bias corrected values of 100% at contrasts below 20% when the hits and correct rejections are lower than 100%. Please explain. Also, the same Figures show normalized detection thresholds. I assume that these were normalized to the detection threshold for the visual-only condition, but what do the error bars then represent for that condition (since all normalized thresholds for the visual-only condition are 1)? Please clarify. How was d’ computed (to obtain criterion bias values) when correct rejections were 100% (or false alarms 0%)? Please provide more details. Why was the β (steepness) parameter fixed for the optostim and visual-only conditions when fitting the psychometric curves?

2. I applaud the use of the control in which the visual stimulus was presented at the GCaMP-only site. Were these tests done interleaved with the sessions in which the C1V1-site was stimulated with the visual target? If the control measurements were run after the C1V1 site experiments, can the authors then exclude that the smaller behavioral effects were because the monkeys learned to ignore or were less distracted – to some extent – by the optostimulation?

3. The authors should provide the size of the fixation window.

4. A statistical clarification: the authors write that "Statistical comparisons in Figures 3 and 4 are made between bootstrapped distributions, using two-tailed, unpaired Student's t-test.". But the bootstrapped distributions are distributions of means and thus how can one employ than a Student-t test?

5. Supplementary Figure 5: I am confused about the legend and what is represented in this Figure: are blank trials meant to be visual-only trials?

*Reviewer #2:*

This study is commendable for the high degree of technical difficulty in executing the experiments to demonstrate an all optical interface to both record and modulate neural activity in awake, behaving non-human primates. They combine widefield fluorescence imaging and optogenetic stimulation while monkeys perform a visual task. The authors claim that their use of low opto power is novel, but this has been done at least once before in monkeys (Ju et al., 2018; ref [4] in the manuscript; a direct comparison with Ju et al. 2018 would be helpful).

The authors ask two main questions. First, whether low-power optogenetic stimulation ("optostim") of primary visual cortex will produce a psychophysical masking effect similar to a visual mask of increased background luminance. Second, they ask whether the interaction between the optostim and the visual stimulus interact sublinearly, consistent with contrast responses in V1.

Monkeys performed a visual detection task while optogenetic stimulation and optical read-out of neural activity was performed, and were rewarded for correctly detecting the visual stimulus, regardless of the optogenetic stimulation condition. On optostim trial blocks, animals had to report the trials in which a visual stimulus was present. Optostim was applied in blocks of trials, rather in a randomized manner. The main findings of the study are that indeed, activation of a population of neurons in V1 does interfere with perceptual reports of a visual stimulus and that stimulus responses are reduced in the presence of optogenetic stimulation. The strong behavioral effect produced using optogenetics in NHPs is important and novel, and has generally been difficult to achieve. This aspect, however, is unconvincing and, unfortunately, buried under the less novel finding of sublinear combination of visual and optogenetic stimulation.

While technically adventurous, the composition of this manuscript is problematic on several scales ranging from the writing to the interpretation of data and the absence of critical information to support their conclusions. Structurally, I am unsure whether this paper is intended to be a methodological description of a novel "toolkit" in NHPs or the report of a scientific finding. It seems to walk the line between both genres, but leaves out crucial information necessary for the detailed understanding of each aspect. The authors might consider writing this as two manuscripts – one that describes the novel toolkit in detail (and how it compares with Ju et al. 2018), which would be helpful to anyone trying to implement it, and another that describes the findings, building upon the group's recent study of the masking effects of natural backgrounds (Bai et al., 2021). Most importantly, however, there are several missing pieces of information that are necessary to evaluate whether the study's conclusions are adequately supported by the data.

1. Why is low-power stimulation used? The rationale is not clearly explained at the start of the paper, but appears to be due to the fact that the wavelength used for GCaMP excitation can also activate C1V1 – the "read" channel is also "writing". To reduce this crosstalk, the light power was reduced from about 1mW/mm^2 to 0.01mW/mm^2, deduced from pilot experiments.

"Pilot detection experiments revealed that, due to the broad excitation spectra of C1V1 [14], the blue GCaMP excitation light can affect the monkey's detection performance. We therefore minimized the blue light level to ~0.01mW/mm2 by running the camera at a low frame rate (20 Hz) and at very low saturation level (see Methods). At this low light level, we did not observe any perceptual effects of blue light illumination… Therefore, under the conditions tested here, the cross-talk between the optical "read" and optical "write" components in our system is negligible."

This is a critical issue, and it is unfortunate the authors do not include any supplementary figures showing the results of these pilot experiments used for the power titration. For this to be a viable tool, it is imperative to demonstrate that the 'read' channel is not also inadvertently 'writing', and to show the threshold at which this is no longer an issue. The authors also need to show (or at the very least quantify) how the behavioral detection performance was affected prior to the power reduction.

2. The authors conclude that "Concurrent optostim and GCaMP imaging revealed that the decline in behavioral detection sensitivity could be attributed to sublinear interaction between the optogenetically and visually driven neural responses in V1; i.e., responses evoked by the visual target decrease significantly when riding on top of an optostim-driven response pedestal."

Does the optogenetic stimulation produce a visual percept, similar to electrical microstimulation of V1? An alternative explanation that cannot be ruled out based on the provided information is that the optogenetic stimulation is producing a phosphene (a visual percept due to the direct stimulation of cortex), and that interferes with perceiving the visual stimulus. The perceptual impact of optogenetic stimulation can be inferred by the false alarm rate associated with optogenetic stimulation in the absence of a visual stimulus (0% contrast). Unfortunately, the authors never show or quantify the false alarm rates for either animal. These data points are also puzzlingly absent from the psychometric curves for contrast 0%. On page 12, the authors write "In this monkey (monkey T), optostim caused mainly an increase in the false alarm rate and a smaller drop in hit rate, and a significant reduction in the monkey's detection criterion." This increase in false alarm rate suggests the monkeys are seeing something in addition to the visual stimulus that could be confusing or distracting. Further, there is no comparison of the saccade trajectories between the optostim and visual stimulus only trials. This could provide some clue as whether/how the optogenetic stimulation is subjectively perceived by the monkey.

I don't see it as a problem if the data suggest that optogenetic stimulation induces a visual percept – this is quite interesting in fact – but it does pose a problem for the authors claim that the change in detection performance is attributable to the reduced stimulus response in the optostim condition. This issue is further complicated by the results of stimulating the GCaMP-only site (Figure 4), which also seems to show significant changes in detection thresholds (Figure 4D,I). This suggests that distraction/confusion could underlie some of the behavioral effects at the C1V1 site as well. To substantiate the argument that the behavioral effects are spatially specific, the authors should directly compare the threshold and criterion changes between the C1V1 and GCaMP-only sites. If the data suggests a phosphene, it would be helpful to attempt to delineate the possible size of such a percept given the receptive field boundaries of the stimulated sites, and then to compare it to the size and location of the visual stimulus.

Particularly since the authors are interested in the implications for neural prosthetics, there should be a more nuanced and careful consideration of all the possible reasons for the changes in detection performance, including evidence for and against the possibility of a phosphene. It would also be useful to see the raw behavioral data to better gauge the strength of the detection impairment the effect of the bias adjustment (i.e.Figure 3C,H).

3. The viral injection methods are inadequately described. What was the total volume of virus and spatial arrangement of injections? This is needed for approximating how many neurons are being activated by the stimulation to produce the psychophysical mask.

4. Similarly, basic stimulus response properties (receptive field location, optimal stimulus properties) of the activation and control sites are minimally described but critical to interpreting the results. It would be helpful to the reader to make these properties as obvious as possible, particularly since there are substantial differences in the individual monkeys' behavior that might be attributable to these factors.

5. Surprisingly, there appears to be no direct comparison of the optostim "mask" to a visual luminance mask. Since the scientific question here is whether optostim can produce the same effects as a visual mask, it seems fundamental to compare the behavioral and neural changes produced by the optostim with a visual pedestal. This comparison would also help elucidate whether the sublinear interactions produced by the visual and optostim are generated by similar mechanisms as that of the visual stimulus pedestal.

[Editors' note: further revisions were suggested prior to acceptance, as described below.]

Thank you for resubmitting your work entitled "Similar neural and perceptual masking effects of low-power optogenetic stimulation in primate V1" for further consideration by *eLife*. Your revised article has been evaluated by Joshua Gold (Senior and Reviewing Editor).

The manuscript has been improved but there are some remaining issues that need to be addressed, as outlined below:

I am including the full comments from two reviewers, below, because they will hopefully be of use for this round of revisions. You'll see that the first reviewer is satisfied with the revisions (although their comments suggest that you might try to more clearly describe how your results relate to past studies).

The second reviewer raised more substantive points. Point 1 involves adding details that I assume will be straightforward. Several of the other points suggest new control behavioral measurements. It seems worth considering seriously the benefit/cost ratio of these measurements -- the reviewer obviously thinks they are worth doing, but I also think that perhaps citing other studies and addressing these concerns more directly in the paper could suffice.

*Reviewer 1:*

I have reviewed the rebuttal of the authors and the revised manuscript. I am satisfied with their replies to my comments and the revisions they made to the manuscript. There is still the somewhat puzzling result that the effects of combined optostimulation and visual stimulation have a different effect on the behavioral choices of the two animals: one animal shows a decrease in hit rate while the other animal an increase in false alarm rate. Both effects result in a decrease in percent correct with optostimulation. The authors argue that the difference in behavioral effect between the animals is due to the different criterion levels without optostimulation, but I am not sure whether that can explain all.

As in my initial review, I believe that the study is from a technical point of view impressive, but its theoretical impact is less since other studies, e.g. Nassi et al., Neuron, 2015, have already shown sub additive effects of optostimulation in the visual cortex. The present study shows a similar effect at lower power densities and provides also behavioral data.

*Reviewer 2:*

First off, I must say that this was an unnecessarily difficult revision to review, mostly due to figure labeling. The response letter figure labels (i.e., "Figure S8") do not match the labels in the revised manuscript (i.e., "Figure 6-supplement 2", etc.). In the end I was left to count figure legends in order to decipher which figure the authors were referring to in the letter. This may seem minor, but it was very time-consuming.

The authors have improved somewhat their manuscript with this revision. The descriptions of the animals' behavioral responses and methodological details are improved, and replacing "pedestal" with "mask" was a good choice. The additions to the discussion are also helpful.

More importantly, however, the revision does not adequately address several of my original major concerns. For instance, I understand that the original recording chambers are no longer available, however, at least 3 concerns (#2-4 below) could have been addressed with purely behavioral experiments. Disappointingly, the authors did not attempt to perform these simpler experiments, or even cite comparable studies to substantiate their conclusions. Overall, I believe that while their study is technically impressive, it is not particularly novel for the broad readership of *eLife*. There is a large amount of overlap with previous studies (Ju et al. 2018, Nassi et al. 2015), as originally pointed out. This study could greatly benefit from additional analysis to unravel the sublinear interactions mechanistically. As it stands, I cannot be supportive of this manuscript.

1) Overall comment 1 response: Figure 6 —figure supplement 2: I appreciate that the authors conducted another experiment to address overall comment 1. However, there is insufficient information to allow for an adequate interpretation of this figure. For example, they do not mention how this recording was performed, how many cells are included in this analysis (or is it just 1 multi-unit response?), how many trials were averaged, does the depth of the electrode correspond to the area directly stimulated by the light, etc., etc.? Was there a simultaneous GCaMP recording? Without these details the figure raises more questions than it answers.

2) Overall comment 2. Show the results of the neural and behavioral data for the power titration experiments. The authors show one additional behavioral plot, with no error estimate. They say: "we have not done a systematic comparison of performance with and without blue light. Our impression was that the monkeys' behavioral thresholds with the low-level blue light were comparable to their thresholds during training without blue light. While we cannot rule out that the blue light had some effect on the monkeys' performance, if such effect existed, it was small." This is inappropriate. I appreciate that the chambers are no longer available, but the authors could have performed purely behavioral experiments in the same animals showing that in the absence of blue light, their detection performance is equal to that of low power blue light (without optostim). The authors need to show evidence that blue light alone does not affect performance ("…, if such effect existed, it was small.").

3) Overall comment 6. Address more directly the nature of the visual percept produced by the optogenetic stimulation. The authors now emphasize in the discussion that a distinguishing feature of their study is that animals were only rewarded for detecting a visual stimulus, and not the optogenetic stimulation of V1 itself. However, this aspect is not unique to their study. Rather given Monkey T's unique behavior on the task (see point 5 below), the authors probably could have made a more direct comparison between the optostim and a visual mask.

4) Overall comment 7: regarding comparing their optostim results with a visual mask was not addressed. It is unfortunate that the authors did not perform an additional behavioral control experiment using a visual mask. In their reply they mention the discussion paragraph "Perceptual consequences of optogenetic stimulation in V1", but there is no mention other studies that used a visual mask and compare the behavioral results. This missing, obvious comparison is surprising given that one of their stated experimental questions is "(1) Can we substitute a visual mask with low-power ((<1 mW/mm2) direct optostim of the visual cortex…".

5) The asymmetry in behavioral responses between the two animals (one shows an increase in false alarms, the other a decrease in the hit rate) does stand out more in this version of the manuscript. The authors' interpretation, attributing this to differences in criterion levels across monkeys, makes sense and seems sufficient to explain the asymmetry. The main problem is that Monkey T has an unstable criterion across different experimental blocks. There are more false alarms on low vs. high contrast control trial blocks suggesting that he was adjusting his criterion level across the different trial blocks to closer match the visual stimulus (smart monkey). This problem could have been avoided had the authors chosen to randomize trials with visual contrasts, rather than presenting individual contrasts in blocks.

6) Page 14, line 272 – this sentence is incomplete: "the monkeys do not seem to be able to 272 learn to compensate for the presence of the optogenetic (Figure 2-S2)."

---

## [Author Response]

Essential revisions:1. There were concerns about your interpretation of your results in terms of "a sublinear interaction between visual and optogenetic evoked V1 responses" without knowing/considering the relationship between the calcium signals you measured and the underlying spiking activity. You will see that one reviewer requests additional imaging experiments with simultaneous electrophysiological recordings to measure (a) the linearity of the relationship between the calcium signal and spiking activity and (b) what fraction of cells show network-mediated suppression (as opposed to expected excitation). Upon discussion, the reviewers agreed that such measures would be helpful but are not necessarily crucial. Instead, there may already be information about this particular calcium indicator in the literature, and it was felt that only a small fraction of neurons were likely to be inhibited by the excitatory opsins. It is requested that you consider these factors (particularly the calcium/spiking relationship) more explicitly in your analyses.

We address this issue in three ways.

– First, we added the following text to our discussion:

– We previously showed that widefield GCaMP signals are approximately linearly related to the locally pooled spiking activity in macaque V1 (Seidemann et al. 2016). Therefore, the observed sublinear interaction is likely to reflect a sublinear interaction at the level of V1 spiking activity.

– Second, in the same paragraph we refer to a new supplementary figure (Figure S8) from a pilot electrophysiology experiment that demonstrates strong sublinear interaction between visual and optogenetic evoked multi-unit spiking activity in V1. These preliminary results, which are part of a separate ongoing study, demonstrate clear sublinear responses to visual and optogenetic stimulation at different combinations of contrasts and light power densities and are qualitatively similar to our GCaMP imaging results.

– Third, we further discuss the study of Nassi et al. (Neuron 2015) that used electrophysiology to demonstrate similar sub-linearity in macaque V1 (however using power densities that are one to two order of magnitude higher than the ones used in our study and in our pilot electrophysiology experiments mentioned above).

2. Show the results of the neural and behavioral data for the power titration experiments.

We thank the reviewers for pointing out the need to expand on this important issue. In an initial experiment which we now show as a supplementary figure (Figure S3), we observed a large drop in the monkey’s performance in the presence of excessive blue excitation light (0.11 mW/mm^2^). Therefore, in subsequent experiments we reduced the power density of the blue light, but we have not done a systematic comparison of performance with and without blue light. Our impression was that the monkeys’ behavioral thresholds with the low-level blue light were comparable to their thresholds during training without blue light. While we cannot rule out that the blue light had some effect on the monkeys’ performance, if such effect existed, it was small. We now mention this in the revised manuscript. In addition, we point out that the level of blue light was an order of magnitude lower than the lowest level of orange light used in each animal. Combining this with the lower effectiveness of blue light relative to orange light in activating the red shifted opsin C1V1, the effect of the blue light was likely to be negligible. We do not have neural data to address this issue since we cannot image the GCaMP signals without blue excitation light. We would also like to point out that the chambers that were used for these experiments are no longer available, so unfortunately, we cannot follow up on this issue with additional measurements.

3. Present the behavioral data with and without the bias-correction and include a figure with the false-alarm rates.

We now better explain and justify our methods for analyzing the behavioral data and include a plot of accuracy as a function of contrast w/o bias correction (Figure S4). More generally, we demonstrate that our conclusions do not depend on the specific way in which we analyze the behavioral data. As a side note, the current version of Figure 3B,G contains the raw data and the rates of false alarm can be inferred from it (1 minus correct reject rate).

4. Include details about the receptive field properties of the stimulated neurons.

We now provide additional information regarding location and approximate size of the receptive fields of the stimulated neurons. This information is presented in Methods under *‘*Behavioral Task with Optical Stimulation’. In addition, we added the following sentence at the beginning of the Result section: “The size of the area activated by optogenetic stimulation was ~2mm^2^ at half of the maximal response. This area contains several hundred thousand neurons [19].”

5. Expand the descriptions of the various methodological details (virus injections, behavioral bias-correction methods, etc.).

We now provide all requested details as described in our reply to individual reviewers below.

6. Consider the possibility that the behavioral impairment involves higher-level processing (e.g., where an optogenetically induced percept interferes with visual processing).

To address this issue, we added a new section to the discussion: ‘Perceptual consequences of optogenetic stimulation in V1’. Briefly, the focus of our study is on the interactions between visual and optogenetic stimulation in V1 and we deliberately designed our behavioral task so that it would depend solely on the visual stimulus and would be insensitive to the exact nature of the percept evoked by the optogenetic stimulation on its own. Our results reveal that optogenetic stimulation of V1 is altering the monkeys’ perception of the visual target in a way that is consistent with visual masking. Similar to visual masking, this effect is spatially selective, and the monkeys do not seem to be able to learn to compensate for the presence of the optogenetic stimulation even though they are incentivized to do so (we now include a new supplementary figure (Figure S2) – to demonstrate this important point). These results are consistent with the physiological reduction in the visual evoked response in the presence of the optogenetic stimulation. Therefore, our results hold irrespective of the exact percept that is elicited by the optogenetic stimulation on its own.

7. Consider addressing more directly the nature of the visual percept produced by the optogenetic stimulation. For example, it was suggested that it would be useful to compare the optogenetically induced mask with a purely visual mask. We are not requiring such additional experiments but raise them as a possibility to consider, because it was thought that they would be extremely helpful in providing an approximate measure for the brightness of the optogenetically induced visual percept.

We believe that additional experiments with a visual mask, while interesting, are unlikely to contribute significantly to the current study. We added a paragraph to the discussion under ‘Perceptual consequences of optogenetic stimulation in V1’ where we explain that we were not attempting to generate an optogenetic-evoked response that is indistinguishable from the response to the visual target. Specifically, we explain that while the visual target was placed so that its center fell at the center of the receptive field of the C1V1-expressing neurons, the visual target most likely activated a larger cortical region than the optogenetically activated area and evoked a response with a different spatial profile. Therefore, optostim masking is similar to visual masking, where the mask does not have to be identical to the target to lead to a perceptual masking effect.

Please see below for more details.Reviewer #1 (Recommendations for the authors):1. Concern regarding the relationship between the GCaMP signal and spiking activity.

This concern was discussed above (Overall #1).

2. Reviewer #1 was concerned that optostimulation could lead to heterogenous effect by suppressing some V1 cells.

We now mention and cite our previous work (Seidemann et al. 2016) that showed that the with the CaMKII promoter we obtain expression in ~90% of excitatory V1 cells. This would suggest that the dominant effect of optostimulation would be to elevated population response, as we observe in our pilot electrophysiological recordings (Figure S8).

3. Reviewer #1 pointed out that the explanation of our bias correction procedure was unclear.

We now include more details of the maximum likelihood approach in our Methods. We clarified how the optimal criterion is defined in our formulation, how 0% and 100% data points were treated, and how the bias corrected values in Figure 3 and 4 were calculated from the fitting.

4. Explain error bars on normalized detection thresholds.

Error bars for the normalized thresholds in Figures3 and 4 indicate the standard deviation of the bootstrapped threshold values divided by the overall mean visual threshold. We have further clarified this in the Methods as well as in the figure caption.

5. How was d' computed (to obtain criterion bias values) when correct rejections were 100% (or false alarms 0%)?

Please see our reply to #3 above.

6. Why was the β (steepness) parameter fixed for the optostim and visual-only conditions when fitting the psychometric curves?

We found that the steepness parameter had overlapping confidence intervals when fitted individually. We have added this explanation to our Methods.

7. Reviewer #1 asked if the GCaMP-only sessions were interleaved with the sessions in which the C1V1-site was stimulated.

The answer is yes. The sequence of experiments is now presented in Figure S2. This figure also shows that the monkeys did not learn to ignore the behavioral effects of optostimulation over time.

8. Report the size of the fixation window.

This is now reported in Methods.

9. A statistical clarification: the authors write that "Statistical comparisons in Figures 3 and 4 are made between bootstrapped distributions, using two-tailed, unpaired Student's t-test.". But the bootstrapped distributions are distributions of means and thus how can one employ than a Student-t test?

We have corrected our statistical calculation and updated its description in the Statistics section of our Methods.

10. Supplementary Figure 5: I am confused about the legend and what is represented in this Figure: are blank trials meant to be visual-only trials?

This figure is now Figure S9. The purpose of this figure is to compare pulsed and continuous optogenetic stimulation. The results suggest that the evoked response is relatively independent of the optostim temporal modulation (i.e., the response mainly depends on the time average of the optostim power). The legend of the figure has been revised and hopefully it is clearer.

Reviewer #2 (Recommendations for the authors):1. A direct comparison with Ju et al. 2018 would be helpful.

Ju et al. is mentioned in several points in the paper. We now explicitly mention in the discussion that Ju et al. 2018 were also able to obtain a behavioral effect at relatively low power optogenetic stimulation though they did not employ an optical readout during this portion of their work. We also mention in the discussion Ju et al. as an example of a study in which the animal was reward to report the presence of optogenetic stimulation, which contrasts with our approach in which the animal was only rewarded based on the visual stimulus.

2. Reviewer #2 was unsure whether this paper is intended to be a methodological description of a novel "toolkit" in NHPs or the report of a scientific finding.

Our paper addresses important scientific questions regarding nonlinear computations in V1 and their impact on perception using a new toolkit. Our intention was not to write a technical paper. However, our goal is to provide all the technical details necessary in order to replicate our study and we believe that the revised version achieves this goal.

3. Why is low-power stimulation used? The rationale is not clearly explained at the start of the paper, but appears to be due to the fact that the wavelength used for GCaMP excitation can also activate C1V1 – the "read" channel is also "writing". To reduce this crosstalk, the light power was reduced from about 1mW/mm^2 to 0.01mW/mm^2, deduced from pilot experiments.

We apologize for the confusion. When we mention low-power stimulation we refer to the low power density of orange light used for optogenetics. This is in contrast to previous studies that required orders of magnitude higher power densities, which could cause heating and damage. Because we obtain high levels of expression and sensitivity to optostimulation, we had to reduce the power density of the blue light used for exciting the GCaMP. We now explain this more clearly in the paper.

4. Reviewer #2 wanted better description of the pilot experiments that demonstrated a behavioral effect of the blue light.

This is discussed in our reply to Overall point #2.

5. Reviewer #2 was concerned about the possibility that optogenetic stimulation elicited a perception of a phosphene.

This is discussed in our reply to Overall point #6.

6. The reviewer was concerned that the behavioral effects when the visual target was in the GCaMP only site may reflect non-selective distraction/confusion.

We now mention in the discussion that while the visual stimuli used for the C1V1 sites and the GCaMP only sites did not overlap in the visual field, the population receptive fields at these two locations partially overlapped, so that the visual target at one location partially activated some of the neurons at the other location. This could explain the small behavioral effect of optostim when the target was presented at the location corresponding to the GCaMP only site. However, the finding that despite the proximity of these visual stimuli, the behavioral effects are drastically smaller when the visual stimulus was at the location corresponding to the GCaMP only site, strongly suggest that this effect is due to visual masking rather than general confusion.

7. The perceptual impact of optogenetic stimulation can be inferred by the false alarm rate associated with optogenetic stimulation in the absence of a visual stimulus (0% contrast). Unfortunately, the authors never show or quantify the false alarm rates for either animal. These data points are also puzzlingly absent from the psychometric curves for contrast 0%. On page 12, the authors write "In this monkey (monkey T), optostim caused mainly an increase in the false alarm rate and a smaller drop in hit rate, and a significant reduction in the monkey's detection criterion." This increase in false alarm rate suggests the monkeys are seeing something in addition to the visual stimulus that could be confusing or distracting.

We apologize for our lack of clarity. We now explain more clearly that panels B and G in Figure 3 report the percent correct reject, and one minus this value is monkey’s false alarm rate. Monkey L had a high criterion, made few false alarms and the main effect of optostim in this monkey was to cause a drop in the hit rate. Monkey T had a more moderate criterion, and in this monkey the main effect of optostim was to cause an increase in false alarm rate. Once we correct for the animal’s bias, optostim had similar effect of reducing both monkey’s sensitivity. We disagree with the reviewer’s assertion that “the perceptual impact of optogenetic stimulation can be inferred by the false alarm rate associated with optogenetic stimulation in the absence of a visual stimulus”. The two monkeys had a very different false alarm rated in the absence of optogenetic stimulation, so false alarm rate can reflect the animal’s criterion rather than what it perceives. Similarly, even if the monkey perceived a phosphene, it could have easily adjusted its criterion so that the false alarm rate would have remained the same as without optogenetic stimulation. This is the reason that we focus on the effect of optostim on the animal’s sensitivity.

8. It would also be useful to see the raw behavioral data to better gauge the strength of the detection impairment the effect of the bias adjustment (i.e.Figure 3C,H).

As discussed above (Overall #3), we now include a version of Figure 3C,H with no bias correction (Figure S4). However, we would like to point out that Figure 3 does contain the raw data in panels A,B,F,G, and that the bias corrected psychometric curves in Figure 3C,H are the appropriate way to assess the effect of optostim on the animal’s behavioral sensitivity.

9. Reviewer #2 suggested that we include comparison of the saccade trajectories between the optostim and visual stimulus only trials.

This is a good suggestion. We now include such an analysis in Figure S6. The analysis reveals no systematic differences in eye movements between optostimulation and visual only trials.

10. Viral injection methods are inadequately described.

We now provide in Methods details about the volume of virus, the arrangement of the injections and the approximate area activated by optogenetic stimulation.

11. Basic stimulus response properties (receptive field location, optimal stimulus properties) of the activation and control sites are minimally described.

We provide details regarding the location of the receptive fields and their approximate size in the legends of Figure 2 and Figure S1 and in Methods. We also added a paragraph in the discussion under ‘Perceptual consequences of optogenetic stimulation in V1’ where we describe that it is likely that optostim evoked a response in a cortical area that is smaller than the response evoked by the visual target.

12. The reviewer suggested that it would be useful to compare the optogenetically induced mask with a purely visual mask.

Please see our reply to comment #7.

[Editors' note: further revisions were suggested prior to acceptance, as described below.]

Reviewer 1:I have reviewed the rebuttal of the authors and the revised manuscript. I am satisfied with their replies to my comments and the revisions they made to the manuscript. There is still the somewhat puzzling result that the effects of combined optostimulation and visual stimulation have a different effect on the behavioral choices of the two animals: one animal shows a decrease in hit rate while the other animal an increase in false alarm rate. Both effects result in a decrease in percent correct with optostimulation. The authors argue that the difference in behavioral effect between the animals is due to the different criterion levels without optostimulation, but I am not sure whether that can explain all.

The first comment is a minor concern related to the difference between the behavioral effect of optogenetic stimulation in the two monkeys. If optogenetic stimulation acts as a visual mask, it should lead to a decrease in sensitivity (d’) with respect to the visual target, as observed in both monkeys. But there is no a priory reason to expect that a mask would have the same effect on different subjects’ criterion. Reviewer #1 writes that s/he is not sure that a difference in criterion can explain all of the difference between the two monkeys, but does not offer any alternative interpretation that we could test or explain how an alternative explanation could affect our conclusions. We would also like to point out that reviewer #2 states that our interpretations of the differences in the behavioral effects of optostimulation between the two monkeys makes sense.

As in my initial review, I believe that the study is from a technical point of view impressive, but its theoretical impact is less since other studies, e.g. Nassi et al., Neuron, 2015, have already shown sub additive effects of optostimulation in the visual cortex. The present study shows a similar effect at lower power densities and provides also behavioral data.

The second comment is that our study is impressive technically but has a limited theoretical impact because of the prior finding of Nassi et al. from 2015 demonstrating sublinear interactions between optogenetic and visual stimulation in fixating macaque V1. We now explain more explicitly the unique contributions of our study. First, we add to the discussion the following text with respect to the Nassi et al. paper: “in that study, the light power densities required to elicit optogenetic-evoked responses were one to two orders of magnitude higher than in the current study. Such high-power densities raise concerns regarding tissue heating, which could affect the nature of the interactions between visual and optogenetic-evoked responses in V1. Therefore, our results, in addition to demonstrating a novel perceptual interaction between visual and optogenetic stimulation in V1, significantly extend the results of Nassi et al., and provides an important confirmation of their main finding.” In addition, in the opening sentence of the discussion, we further emphasize the unique aspects of our work. “Here we used our novel optical-genetic toolkit for all-optical bi-directional probing of macaque cortex to study simultaneously, for the first time, the neural and perceptual masking effects of optogenetic stimulation in macaque V1.” Finally, we further emphasize in the discussion that our study is the first to directly test perceptual substitution of a visual stimulus by optogenetic stimulation in V1. We believe that these are highly innovative and important contributions to the field.

Reviewer 2:First off, I must say that this was an unnecessarily difficult revision to review, mostly due to figure labeling. The response letter figure labels (i.e., "Figure S8") do not match the labels in the revised manuscript (i.e., "Figure 6-supplement 2", etc.). In the end I was left to count figure legends in order to decipher which figure the authors were referring to in the letter. This may seem minor, but it was very time-consuming.

First, we would like to apologize for the inconsistent supplementary-figure labeling. We originally had the supplementary figures labeled Figure S1-S8. Then, when we received the request to change the numbers to *eLife*’s supplementary figures format, we neglected to change the numbers in our reply.

The authors have improved somewhat their manuscript with this revision. The descriptions of the animals' behavioral responses and methodological details are improved, and replacing "pedestal" with "mask" was a good choice. The additions to the discussion are also helpful.More importantly, however, the revision does not adequately address several of my original major concerns. For instance, I understand that the original recording chambers are no longer available, however, at least 3 concerns (#2-4 below) could have been addressed with purely behavioral experiments. Disappointingly, the authors did not attempt to perform these simpler experiments, or even cite comparable studies to substantiate their conclusions. Overall, I believe that while their study is technically impressive, it is not particularly novel for the broad readership of eLife. There is a large amount of overlap with previous studies (Ju et al. 2018, Nassi et al. 2015), as originally pointed out. This study could greatly benefit from additional analysis to unravel the sublinear interactions mechanistically. As it stands, I cannot be supportive of this manuscript.

Second, the reviewer argues that “There is a large amount of overlap with previous studies (Ju et al. 2018, Nassi et al. 2015).” We strongly disagree with this assessment. As described in our reply to point 2 of reviewer #1 and in the manuscript, our study goes well beyond the study Nassi et al. in multiple important ways. While Ju et al. 2018 have shown neural and behavioral effects of low-power optogenetic stimulation, there are multiple key differences between their study and ours. (1) Our study is the first to examine the perceptual interaction between visual and optogenetic stimulation and to examine the parallels between visual and optogenetic masking. (2) In their study, the monkey was rewarded for reporting the optogenetic stimulation. As we explain in our manuscript, such a design does not test for perceptual substitution, and therefore our study is the first to demonstrate clear perceptual substitution with optogenetics. (3) Ju et al. did not combine simultaneous imaging, optogenetic stimulation and behavior, and the neural and behavioral effects of optogenetic stimulation were studies under very different regimes. We therefore believe that our study provides multiple novel contributions that go well beyond previous studies.

1) Overall comment 1 response: Figure 6 —figure supplement 2: I appreciate that the authors conducted another experiment to address overall comment 1. However, there is insufficient information to allow for an adequate interpretation of this figure. For example, they do not mention how this recording was performed, how many cells are included in this analysis (or is it just 1 multi-unit response?), how many trials were averaged, does the depth of the electrode correspond to the area directly stimulated by the light, etc., etc.? Was there a simultaneous GCaMP recording? Without these details the figure raises more questions than it answers.

The reviewer points out some details that are missing from the legend of a new supplementary figure (Figure 6-supplement 2). This figure shows multi-unit responses from a single recording session taken at a depth of ~300 microns (well within the range of depths contributing to the GCaMP signals) at the center of the C1V1 site in monkey L. The responses are averages over 10 repeats. This figure was added in order to reassure this reviewer that the sublinear interaction that we observe is not unique to the GCaMP signal. However, given the very preliminary nature of this figure, we now believe that it would be better not to include it in the current manuscript so we eliminated this figure from the revised version.

2) Overall comment 2. Show the results of the neural and behavioral data for the power titration experiments. The authors show one additional behavioral plot, with no error estimate. They say: "we have not done a systematic comparison of performance with and without blue light. Our impression was that the monkeys' behavioral thresholds with the low-level blue light were comparable to their thresholds during training without blue light. While we cannot rule out that the blue light had some effect on the monkeys' performance, if such effect existed, it was small." This is inappropriate. I appreciate that the chambers are no longer available, but the authors could have performed purely behavioral experiments in the same animals showing that in the absence of blue light, their detection performance is equal to that of low power blue light (without optostim). The authors need to show evidence that blue light alone does not affect performance ("…, if such effect existed, it was small.").

The reviewer is still concerned that the low-power blue excitation light used for GCaMP imaging during our experiments could have had a behavioral effect. At the request of this reviewer, we added a supplementary figure from a pilot experiment showing that high-power blue light can indeed affect behavior. However, we did not systematically examine a possible small behavioral effect of the low-power blue light as this was not important for the current study. This is something that could be addressed in future studies by manipulating the relative titers of the opsin and GCaMP expressing viruses to make the site’s opsin expressing cells less sensitive to blue light stimulation. Importantly, we now add the following sentence to the revised manuscript: “However, because our experiment focuses on the comparison between visual only and visual-plus-optostim trials, and the blue light was present in both trial types, even if it had a small perceptual effect, this has no impact on the conclusions of our study.“

3) Overall comment 6. Address more directly the nature of the visual percept produced by the optogenetic stimulation. The authors now emphasize in the discussion that a distinguishing feature of their study is that animals were only rewarded for detecting a visual stimulus, and not the optogenetic stimulation of V1 itself. However, this aspect is not unique to their study. Rather given Monkey T's unique behavior on the task (see point 5 below), the authors probably could have made a more direct comparison between the optostim and a visual mask.

We emphasized that a distinguishing feature of our study is that animals were only rewarded for detecting a visual stimulus, and not the optogenetic stimulation of V1 itself. The reviewer writes: “However, this aspect is not unique to their study.”, but does not cite any other study, so it is not clear if and now we can address this concern (since we are unaware of any other study).

4) Overall comment 7: regarding comparing their optostim results with a visual mask was not addressed. It is unfortunate that the authors did not perform an additional behavioral control experiment using a visual mask. In their reply they mention the discussion paragraph "Perceptual consequences of optogenetic stimulation in V1", but there is no mention other studies that used a visual mask and compare the behavioral results. This missing, obvious comparison is surprising given that one of their stated experimental questions is "(1) Can we substitute a visual mask with low-power ((<1 mW/mm2) direct optostim of the visual cortex…".

The reviewer complains that we have not done additional behavioral experiments to compare the effect of optostimulation with a visual mask. As we explained in our reply to the original reviews, we believe that additional experiments with a visual mask, while interesting, are unlikely to contribute significantly to the interpretation of our current results and were likely to raise more questions. We are planning to perform such measurements but those would be more appropriately described in a future publication. In the reviewing editor’s summary of the original review, it was mentioned explicitly that such behavioral studies are not a requirement for publication. Further, in our approved action plan, we did not mention additional behavioral experiments. Therefore, we think that it is unfair and unreasonable to bring up this criticism at this stage. The reviewer is also concerned that we did not mention “other studies that used a visual mask and compare the behavioral results.” We are not aware of a human or monkey study that used a similar design that could be directly compared with our results.

5) The asymmetry in behavioral responses between the two animals (one shows an increase in false alarms, the other a decrease in the hit rate) does stand out more in this version of the manuscript. The authors' interpretation, attributing this to differences in criterion levels across monkeys, makes sense and seems sufficient to explain the asymmetry. The main problem is that Monkey T has an unstable criterion across different experimental blocks. There are more false alarms on low vs. high contrast control trial blocks suggesting that he was adjusting his criterion level across the different trial blocks to closer match the visual stimulus (smart monkey). This problem could have been avoided had the authors chosen to randomize trials with visual contrasts, rather than presenting individual contrasts in blocks.

The reviewer was concerned about “Monkey T having an unstable criterion across different experimental blocks. There are more false alarms on low vs. high contrast control trial blocks suggesting that he was adjusting his criterion level across the different trial blocks to closer match the visual stimulus (smart monkey).” As the task becomes harder, the monkey is expected to lower his criterion, which could lead to more false alarms. We chose not to randomize target contrasts within a block because this complicates the task (by introducing target-contrast uncertainty). A design in which there is only one target contrast within a block is standard in human psychophysics. Since the same design was used in visual-only and in optostim blocks, there is no reason to think that it had an impact on our main results.

6) Page 14, line 272 – this sentence is incomplete: "the monkeys do not seem to be able to 272 learn to compensate for the presence of the optogenetic (Figure 2-S2)."

Thank you for pointing out this problem. This has been corrected.